# FedSal: Enhancing Federated Graph Learning Through Saliency Aware Client Clustering

## Abstract

Graph Neural Networks (GNNs) are essential for analyzing structured data but face significant challenges in federated learning (FL) environments, where non-IID client distributions and structural heterogeneity impede convergence and performance. To address these issues, we introduce Federated Saliency Aggregation Learning (FedSal), the first framework to apply saliency maps in GNN-based FL on graph classification tasks. FedSal replaces full-gradient uploads with compact saliency activations, enabling dynamic clustering of clients via simple thresholds ($\epsilon_{\mathrm{mean}}, \epsilon_{\mathrm{max}}$) and cluster-wise model averaging. We further propose FedSal+, which augments node features with positional and random-walk encodings to inject structural priors without exposing raw graph data. Extensive experiments on thirteen molecular, protein, and social-network benchmarks under extreme non-IID splits show that FedSal and FedSal+ achieve higher accuracy, converge faster, and reduce communication cost compared to state-of-the-art methods. These results demonstrate the SOTA performance of saliency-driven clustering for personalized, robust, and communication-efficient federated graph classification tasks.

## 1 Introduction

Graph Neural Networks (GNNs) have emerged as powerful tools for modeling graph-structured data, achieving state-of-the-art performance in domains including molecular property prediction, clinical risk stratification, community detection, and traffic forecasting Wu et al. (2020); Kipf & Welling (2016); Hamilton et al. (2017); Chen et al. (2020); Cui et al. (2019). Traditional GNN training typically aggregates all graph data centrally, an approach often infeasible due to privacy constraints, corporate policies, and scalability issues. Federated Learning (FL) addresses these challenges by exchanging model updates rather than raw data, preserving data privacy and improving scalability McMahan et al. (2016); Kairouz et al. (2021); Li et al. (2020).

Integrating FL with GNNs, termed Federated Graph Neural Networks (FedGNNs), enhances graph learning across heterogeneous, decentralized datasets. However, FedGNNs face significant hurdles, notably the non-independent and identically distributed (non-IID) nature of decentralized data. Local graphs differ substantially in node features, connectivity, and labels, leading to unstable learning and degraded performance Zhao et al. (2018). This complexity necessitates robust methods for personalized and efficient federated training.

Standard FL methods, such as FedAvg McMahan et al. (2016), inadequately address these heterogeneities, as they fail to consider structural and semantic divergences Xie et al. (2021). Clustered FL approaches, which group clients by compatible updates, have shown promise in enhancing personalization and reducing cross-client interference Sattler et al. (2020). For example, Graph Clustered Federated Learning (GCFL) Xie et al. (2021) clusters clients based on raw gradient similarities, yet raw gradients remain inherently noisy, sensitive to scaling, and prone to variance.

To overcome these limitations, we propose **FedSal**, a novel FedGNN framework leveraging saliency activation maps—gradients with respect to model outputs rather than data—to cluster clients effectively. Saliency maps highlight influential features contributing to predictions, providing more stable and representative client summaries compared to raw gradients. Although saliency maps have demonstrated efficacy in interpretability, robustness, and improved performance across various graph tasks Pei et al. (2024); Wang et al. (2024), their application within Federated Graph Neural Networks (FedGNNs) has not previously been explored. FedSal dynamically clusters clients based on semantic

similarity in saliency profiles, enhancing personalized aggregation, model stability, and learning efficiency under extreme non-IID conditions

Specifically, FedSal clusters clients by computing compact, normalized saliency summaries and partitioning a cosine-affinity graph via a Stoer–Wagner minimum-cut algorithm with adaptive thresholds ($\epsilon\_max, \epsilon\_mean$). Aggregating parameters within clusters mitigates conflicts arising from structural divergences and accommodates client-specific preferences. Extending this approach, **FedSal+** incorporates positional and random-walk encodings into saliency maps, injecting structural priors to refine clustering resolution and further improve accuracy.

Our research addresses four central questions: ***RQ1:*** *Do saliency activation profiles better capture inter-client similarities compared to raw-gradient or spectral methods?* ***RQ2:*** *Does injecting positional and random-walk structural encodings into saliency summaries improve cluster coherence and downstream accuracy?* ***RQ3:*** *What is the trade-off between communication overhead (saliency summary size) and convergence performance compared to baselines?* ***RQ4:*** *How do adaptive thresholds* ($\epsilon\_max, \epsilon\_mean$) *influence cluster stability, convergence robustness, and fairness for minority-label clients?*

To address **RQ1** and **RQ4**, FedSal computes normalized saliency summaries each round, partitions these via adaptive minimum-cut clustering, and averages parameters within clusters, ensuring robust handling of dynamic heterogeneity. Addressing **RQ2**, FedSal+ enhances saliency maps with structural priors, leading to finer-grained similarity signals, improved cluster coherence, and higher accuracy across tasks. For **RQ3**, we systematically analyze communication-performance trade-offs, comparing saliency-based methods against gradient- and spectral-based baselines.

Experiments across thirteen benchmarks (seven molecular, three protein, and three social network datasets) under extreme non-IID splits demonstrate that FedSal and FedSal+ consistently surpass baselines in accuracy, convergence speed, and communication efficiency. Ablation studies identify optimal thresholds and cluster counts, highlighting that clients with minority labels benefit significantly from our adaptive protocol.

Our contributions include:

- **Unprecedented Use of Saliency Maps in FL and FedGNNs:** No prior work has employed saliency maps or FedGNNs. This novel application of utilizing saliency maps to identify and cluster clients with similar data feature importance, ensuring that model updates are aggregated more effectively and relevantly. could inspire further research.

- **Introduction of FedSal Architecture:** We propose FedSal, a novel Federated Graph Neural Network (FedGNN) architecture that leverages saliency activation maps for client clustering, enhancing personalization and model performance in FL environments.

- **FedSal+:** Structural-prior augmentation using positional and random-walk encodings to enhance clustering quality and downstream task performance.

- **Superior Performance Over State-of-the-Art FedGNNs:** Robust evaluations on real-world datasets demonstrating substantial improvements over state-of-the-art FedGNN methods.

## 1.1 RELATED WORKS

### 1.1.1 FEDERATED LEARNING

FL was first introduced by McMahan et al. (2016), enabling collaborative training across multiple devices under a central server while preserving data privacy Kairouz et al. (2021); Yang et al. (2019); Lyu et al. (2022); Yu et al. (2025); Liu et al. (2025b); Chen et al. (2025); Dai et al. (2025); Liu et al. (2025a); Tan et al. (2025); Shaikh & Samet (2025); Mai et al. (2024); Fu et al. (2025); Fang et al. (2025); Fu et al. (2024); Gao et al. (2024); Wu et al. (2022); Wang et al. (2022b). One of the most prominent and standard settings for FL is the FedAvg algorithm by McMahan et al. (2017). This algorithm relies on stochastic gradient descent (SGD) based optimization, which affects convergence speed and can lead to unstable learning due to its unbiased estimation and averaging of all model parameters during aggregation at the central server Zhao et al. (2018); Li et al. (2020); Karimireddy et al. (2020). Numerous works have aimed to improve the performance of FedAvg, addressing issues

such as performance in heterogeneous (non-IID) settings Wang et al. (2020); Tan et al. (2022a); Chen et al. (2022a); Zhao et al. (2018); Jeong et al. (2018); Huang et al. (2020), communication speeds Hamer et al. (2020), robustness Wang et al. (2019); Yu et al. (2019); Khaled et al. (2020); Liang et al. (2019); Karimireddy et al. (2020); Li et al. (2020), and generalization ability Hamer et al. (2020). One of the most significant challenges in making FL viable for real-world applications is the non-IID data issue, where clients have heterogeneous labels and feature distributions Luo et al. (2021); Tan et al. (2022b); Chen et al. (2022b). Various approaches have been introduced to address this problem, including clustering techniques, which have demonstrated communication efficiencies through group-level personalization of clients. Methods like model-agnostic meta-learning and personalized FL have shown improved generalizability, reducing the effects of non-IID data Fallah et al. (2020); Chen et al. (2018). A recent trend involves decoupling techniques for better personalization, which have also shown improvements in generalizability T Dinh et al. (2020); Li et al. (2021). However, these methods often come with additional communication overhead, although when paired with clustered FL, they have shown reductions in communication costs.

### 1.1.2 FEDERATED GRAPH NEURAL NETWORKS

FL has matured for image and other Euclidean data, but its adoption for graph data remains nascent. Zhang et al. Zhang et al. (2021a) classify FedGNNs into intra-graph, inter-graph, and graph-structured paradigms, while subsequent taxonomies offer finer distinctions He et al. (2021); Fu et al. (2022); Liu et al. (2024). Intra-graph FedGNNs assign each client a subgraph of a larger network to predict missing nodes Zhang et al. (2021b), infer links Chen et al. (2021), or detect communities Baek et al. (2023), with applications in financial crime detection Suzumura et al. (2019). Graph-structured FedGNNs exploit explicit client relationships—e.g., in personalized image processing Chen et al. (2022c) or traffic forecasting Meng et al. (2021). Inter-graph FedGNNs, our focus, train local GNNs on client-specific graph datasets to boost generalization Xie et al. (2021); Zhu et al. (2022); Jiang et al. (2022); Lou et al. (2021).

Recent work has fused core FL techniques and personalization strategies APPLE Luo & Wu (2022), FedCP Zhang et al. (2023), FedGKD Yao et al. (2024), cluster-aware GCFL Xie et al. (2021), gradient-guided FGSSL Huang et al. (2024), and cross-domain FedSSP Tan et al. (2024)—to improve robustness under severe non-IID graph distributions. Complementary advances include two-channel local training Zheng et al. (2021), dynamic client selection Gu et al. (2023), split-channel learning Tan et al. (2023), latent-link generation Xie et al. (2023), embedding masks for personalization Baek et al. (2023), custom optimization losses Guo et al. (2023), meta-learned hyperparameters Wang et al. (2022a), and hierarchical update clustering Briggs et al. (2020). FPL Huang et al. (2023) introduces cluster-aware prototypes to counter domain shifts, while Wang et al.'s FedSLS Wang et al. (2024) aggregates client updates in a saliency–latent space for image-based FL. In the graph domain, centralized saliency-aware regularization has improved robustness Pei et al. (2024), but no prior work leverages saliency maps for client clustering in a federated setting. Our method fills this gap, demonstrating that saliency-guided similarity can effectively drive clustering and aggregation when the data are graphs rather than images.

## 2 METHODOLOGY

### 2.1 TECHNICAL DESIGN

GNNs are powerful tools for learning graph representations and have been widely used in graph mining applications. The model parameters and gradients of GNNs can reflect the underlying graph structure and feature information. Thus, in the FedSal architecture, GNNs are used as the core model for graph mining within the FL framework.

### 2.1.1 DYNAMIC CLUSTERING WITH SALIENCY AGGREGATION

The FedSal framework introduces a novel approach to dynamically cluster clients by leveraging their transmitted saliency maps. We denote by $S_i(t)$ the aggregated saliency map of client $i$ at round $t$, and define the update $\Delta S_i = S_i(t) - S_i(t-1)$. This mechanism aims to maximize collaboration among homogeneous clients and mitigate the negative effects of heterogeneous clients. When client data distributions are highly heterogeneous, a general FL approach may fail to jointly optimize all local

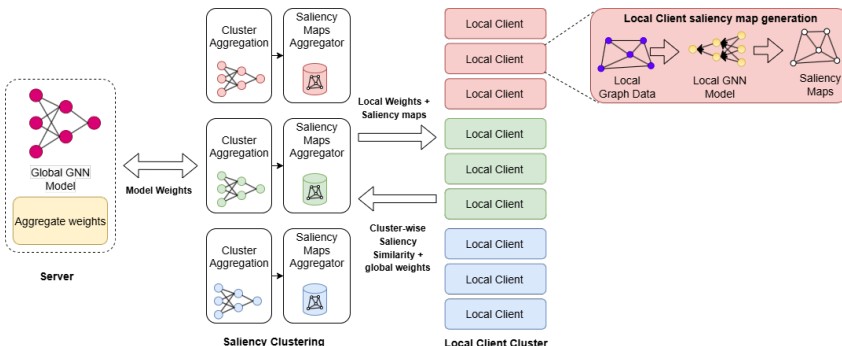

Figure 1: Architectural illustration of FedSal

loss functions. In such cases, after several communication rounds, the general FL algorithm tends to approach a stationary point, indicated by small $\|\Delta S_i\|$ norms. Therefore, clustering is necessary as the FL process nears that point.

To achieve this, we introduce two hyperparameters, $\epsilon_{\text{mean}}$ and $\epsilon_{\text{max}}$, to guide the clustering process:

**Stopping Criterion for General FL:** The mean norm of saliency-map updates across clients is

$$\frac{1}{N} \sum_{i=1}^{N} \|\Delta S_i\| < \epsilon_{\text{mean}},$$

where $N$ is the total number of clients. When this condition holds, the model is considered to be near a stationary point, and further global updates may yield diminishing returns.

**Clustering Criterion:** The maximum norm of saliency-map updates identifies significant heterogeneity:

$$\max_{i=1,\ldots,N} \|\Delta S_i\| > \epsilon_{\text{max}} > 0.$$

A high value indicates at least one client whose data distribution deviates substantially, necessitating client clustering to handle diverse data more effectively.

### 2.1.2 MECHANISM OF CLUSTERING AND AGGREGATION

The FedSal framework employs a top-down bi-partitioning mechanism. At each communication round $t$, the server receives saliency updates $\{\Delta S_i\}$ from clients within an existing cluster $C_k$. If both the mean and maximum update norms exceed the thresholds $\epsilon_{\text{mean}}$ and $\epsilon_{\text{max}}$, the server constructs a cluster-wise cosine similarity matrix $\alpha_k$. Each entry

$$\alpha_{ij} = \frac{\Delta S_i \cdot \Delta S_j}{\|\Delta S_i\| \, \|\Delta S_j\|}$$

serves as the weight of an edge in a fully connected graph whose nodes represent clients in $C_k$. The Stoer–Wagner minimum-cut algorithm is then applied to partition $C_k$ into subclusters $\{C_{k1}, C_{k2}\}$. Further technical details are in Appendix Section 2.1.2.

### 2.1.3 MODEL UPDATE AND AGGREGATION

**Local Update:** Each client $i$ updates its local model $\theta_i(t)$ by minimizing the loss on its local data $D_i$:

$$\theta_i(t) = \arg \min_{\theta} \text{Loss}(\theta; D_i).$$

After the update, the client computes the per-sample saliency

$$S(x) = \left| \frac{\partial \text{Loss}(f(x;\theta),y)}{\partial x} \right|,$$

and aggregates:

$$S_i(t) = \frac{1}{|D_i|} \sum_{x \in D_i} S(x).$$

**Global Aggregation:** For each cluster $k$, the server aggregates client updates:

$$\theta_{t+1,k} = \theta_{t,k} + \sum_{i \in C_k} \Delta\theta_{t,k,i},$$

where

$$\Delta\theta_{t,k,i} = \theta_i(t) - \theta_{t,k}$$

is the update sent by client $i$. Finally, the global model is obtained by averaging across clusters:

$$\theta_{t+1} = \frac{1}{K} \sum_{k=1}^{K} \theta_{t+1,k}.$$

## 2.2 THEORETICAL DESIGN

This section explores the theoretical and practical effectiveness of saliency activation maps in capturing and reducing structural, feature, and task heterogeneity in graph data using GNNs within a clustered FL setting. To substantiate this, we analyze two problem statements and prove them in Appendix Section A.7 through propositions that saliency maps can effectively represent structural, feature, and task information in the model.

**Definition 1** *Saliency Map Distortion in FedSal*: Let $f : \mathcal{G} \to \mathcal{S}$ be a function mapping from the space of GNN parameters $(\mathcal{G}, d)$ to the space of saliency maps $(\mathcal{S}, d')$. The function $f$ is considered to have $\delta$ distortion if for all $u, v \in \mathcal{G}$,

$$\frac{1}{\delta} d(u, v) \leq d'(f(u), f(v)) \leq \delta \, d(u, v).$$

This definition ensures that the relationship between graph parameters and their corresponding saliency maps maintains a bounded distortion, thereby preserving the structural and feature-based relationships even after transformation to the saliency space.

**Theorem 1** *Bourgain's Embedding Theorem*: Bourgain's theorem states that any finite metric space $(X, d)$ can be embedded into Euclidean space with distortion at most $O(\log n)$, where $n$ is the number of points in $X$. Bourgain (1985)

**Problem 1**: FedSal involves the communication of saliency maps between models with heterogeneous graph structures distributed among different clients. The structure and feature differences can be captured by the saliency maps.

**Proposition 1** *Structural Sensitivity of Saliency Maps*: Given a model $M$ with structure represented by the normalized graph Laplacian $L$, features $X$, and saliency map $S$. If we have another model $M'$ with a different structure $L'$, then the saliency-map difference $\|S' - S\|_2$ is bounded by the structural difference.

**Proposition 2** *Feature Sensitivity of Saliency Maps*: Given a model $M$ with structure $L$, features $X$, and saliency map $S$. If we have another model $M'$ with different features $X'$, then the saliency-map difference $\|S' - S\|_2$ is bounded by the feature difference.

**Problem 2**: The communicated saliency maps in FedSal can also capture task heterogeneity.

**Proposition 3** *Task Sensitivity of Saliency Maps*: Given a model $M$ with structure $L$ and features $X$. If trained on different tasks, the resulting models will have saliency maps whose differences are bounded.

## 3 EXPERIMENTS

### 3.1 EXPERIMENTAL CONFIGURATIONS

**Datasets and Partitioning:** We utilize 13 graph classification datasets spanning three domains, as detailed in Appendix Section A.5. These include seven molecule datasets (MUTAG, BZR, COX2,

DHFR, PTC_MR, AIDS, NCI1), three protein datasets (ENZYMES, DD, PROTEINS), and three social network datasets (COLLAB, IMDB-BINARY, IMDB-MULTI). Node features are present in some datasets, and labels are either binary or multi-class in a **graph classification** task.

Two data partitioning settings are used: **Single-dataset setting:** Graphs from a single dataset are randomly distributed among clients, each client receives about 100 graphs, with 10% reserved for testing. **Multi-dataset setting:** Multiple datasets are distributed among clients in groups, with 10% of graphs held for testing.

**Baselines and Model Configuration:** We compare FedSal and FedSal+ with a compact yet representative set of FL baselines, **Selftrain** as the first baseline, the classical aggregators **FedAvg**McMahan et al. (2016) and **FedProx** Li et al. (2018); Yuan & Li (2022), the gradient or cluster-aware methods **GCFL** Xie et al. (2021), the personalised/feature-separation methods **APPLE** Luo & Wu (2022), **FedCP** Zhang et al. (2023), **FedSage** Zhang et al. (2021b), **FGSSL** Huang et al. (2024), Fedstar Tan et al. (2023) and the current cross-domain FedGNN state-of-the-art, **FedSSP** Tan et al. (2024). We omit FedSSP, Fedstar and FedSal+ from IID single-dataset experiments, since their cross-domain cues (domain-adaptive thresholds in FedStar, spectral projections in FedSSP, positional/random-walk encodings in FedSal+) collapse without heterogeneity, and reserve these comparisons for multi-domain benchmarks. The models feature three layers with a hidden size of 64 and are trained using a batch size of 128. An Adam optimizer is employed with a learning rate of 0.001 and a weight decay of $5 \times 10^{-4}$.

**Parameter Settings and Computational Resources:** Local epochs are set to 1 for all methods. Clustering hyperparameters $\epsilon$mean and $\epsilon$max are adjusted based on a grid search-based selection for the best performing values. Experiments are conducted on a server with an NVIDIA RTX 4050 GPU, with 16GB of memory, and all experiments are repeated five times for statistical validation.

**Enhancement of FedSal with FedStar:** Inspired by the FedStar framework Tan et al. (2023), we developed **FedSal+**, integrating advanced encoding strategies such as positional and random walk encodings to enhance structural details in FL clients. These enhancements aim to refine client-side feature representations, improving FedSal's saliency-based model aggregation. By enriching the feature set for each node, FedSal+ produces more detailed saliency maps, focusing on pivotal features during FL, and tests FedSal's compatibility with methods like structural embeddings. Implementation details are provided in Appendix Section A.4. The primary aim of FedSal+ is to determine if the enriched feature set can enhance FedSal's effectiveness, particularly in handling non-IID data in multi-dataset federated settings. We hypothesize that these refined feature representations will lead to more precise and informative saliency maps, thereby improving overall model performance in multi-dataset federated environments. This hypothesis sets the stage for our experimental investigations to validate the improved performance metrics that FedSal+ could deliver through advanced structural embeddings.

## 3.2 TEST ACCURACY ANALYSIS

Table 1: Test accuracy (%) results across different FL Methods (mean ± standard deviation)

| Dataset (# of Clients) | Multi-Dataset | | | Single-Dataset | | |
|---|---|---|---|---|---|---|
| | **Molecules (7)** | **Biochem (8)** | **Mix (13)** | **IMDB-BINARY (10)** | **NCI1 (30)** | **PROTEINS (10)** |
| Selftrain | 75.35±0.45 | 70.53±0.48 | 69.89±0.39 | 76.86±3.72 | 62.42±1.60 | 73.85±1.28 |
| FedProx | 73.80 ± 0.65 | 70.20 ± 1.10 | 66.00 ± 0.90 | 75.20 ± 2.90 | 64.20 ± 1.60 | 72.00 ± 1.20 |
| FedAvg | 75.37±1.21 | 69.49±0.58 | 69.25±0.77 | 77.59±2.45 | 66.12±1.33 | 74.65±1.18 |
| FedSage | 75.90 ± 0.55 | 72.40 ± 0.90 | 68.00 ± 0.60 | 77.80 ± 2.40 | 66.50 ± 1.30 | 74.80 ± 1.10 |
| GCFL | 76.08±0.64 | 70.94±0.88 | 69.10±0.73 | 78.19±2.32 | 65.24±2.28 | 75.19±1.74 |
| APPLE | 76.40 ± 0.70 | 71.60 ± 0.75 | 68.10 ± 0.80 | 76.50 ± 2.60 | 64.40 ± 1.30 | 74.50 ± 1.15 |
| FGSSL | 76.60 ± 0.60 | 72.80 ± 0.70 | 71.05 ± 0.50 | 77.55 ± 2.20 | 64.70 ± 1.40 | 74.90 ± 1.10 |
| FedCP | 78.40 ± 0.88 | 73.10 ± 0.80 | 70.95 ± 0.70 | 78.00 ± 2.50 | 65.00 ± 1.50 | 75.00 ± 1.20 |
| **FedSal** | 76.61 ± 1.03 | **73.46 ± 0.72** | **71.49 ± 0.52** | **80.22 ± 2.63** | **69.91 ± 1.37** | **75.19 ± 0.75** |
| FedStar | 77.02 ± 0.39 | 70.30 ± 0.61 | 68.90 ± 0.89 | – | – | – |
| FedSSP | 78.22 ± 0.70 | 72.03 ± 0.65 | 69.55 ± 0.55 | – | – | – |
| **FedSal+** | **79.87 ± 0.86** | **74.37 ± 0.44** | **72.07 ± 0.61** | – | – | – |

From the test accuracy results in Table 1. In the single-dataset experiments, FedSal consistently outperforms all competing methods like classical aggregators, personalized approaches and self-training alike, while also exhibiting the most stable performance across clients. In the multi-dataset benchmarks, FedSal+ takes the lead across every setting, clearly beating both general-purpose FL algorithms and state-of-the-art cross-domain techniques such as FedSSP and FedStar. Traditional methods like FedAvg and FedProx struggle under heterogeneous graph distributions, and even specialized solutions (e.g., FedStar, FedCP) cannot match the robustness and scalability delivered by our saliency-based schemes. These trends demonstrate that saliency-driven aggregation not only elevates overall accuracy but also adapts more effectively to cross-dataset heterogeneity than any of the existing baselines.

Table 2: Communication Overhead Results (avg communication time per round in seconds)

| Method | Multi-Dataset | | | Single-Dataset | | |
| | Molecules (7) | Biochem (8) | Mix (13) | IMDB-BINARY (10) | NCI1 (30) | PROTEINS (10) |
| --- | --- | --- | --- | --- | --- | --- |
| FedAvg | 0.68 | 1.07 | 2.91 | 0.14 | 0.49 | 0.16 |
| GCFL | 0.71 | 1.12 | 3.03 | 0.20 | 0.84 | 0.21 |
| FedCP | 0.60 | 0.95 | 2.45 | 0.12 | 0.42 | 0.14 |
| FedProx | 0.70 | 1.09 | 3.00 | 0.15 | 0.50 | 0.17 |
| FedSage | 0.73 | 1.18 | 3.20 | 0.16 | 0.52 | 0.18 |
| APPLE | 0.75 | 1.25 | 3.35 | 0.17 | 0.55 | 0.19 |
| FGSSL | 0.85 | 1.45 | 4.10 | 0.19 | 0.65 | 0.21 |
| **FedSal** | 1.12 | 1.78 | 4.73 | 0.23 | 0.86 | 0.26 |
| FedStar | 0.88 | 1.70 | 6.70 | – | – | – |
| FedSSP | 2.25 | 4.30 | 15.50 | – | – | – |
| **FedSal+** | 1.45 | 2.79 | 11.69 | – | – | – |

**Communication Overhead Analysis:** While prior FedGNN studies have not acknowledged communication efficiency in favor of accuracy, we measure average per-round communication time (Table 2) to highlight its importance. Both FedSal and its enhanced variant, FedSal+, incur higher latency than lightweight baselines such as FedAvg, and this gap widens on larger, more heterogeneous datasets. Nevertheless, even at this elevated cost, they still communicate faster than state-of-the-art FedGNN baselines (e.g., FedSSP), preserving a practical edge when bandwidth or response time is critical. Thus, practitioners can attain the accuracy gains of saliency-based aggregation without bearing the highest communication burden among current methods.

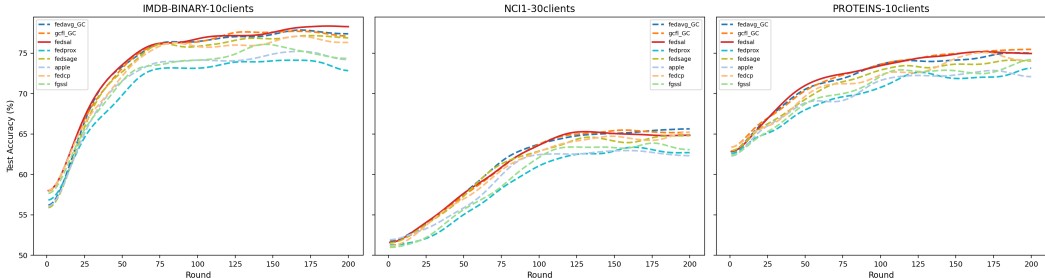

Figure 2: Convergence speed on IMDB-BINARY, NCI1-30 and PROTEINS-10 datasets (left to right).

**Analysis of Convergence Speed:** The convergence speed and efficiency of the models are illustrated in Table 3 and Figures 2 and 3. Across single-dataset scenarios, FedSal converges at essentially the same accuracy as mainstream baselines such as FedAvg and GCFL, posting only marginal but consistent gains. In contrast, the advantages of saliency-aware aggregation become clear once cross-dataset heterogeneity is introduced. On the multi-dataset benchmarks, both FedSal and its enhanced variant FedSal+ lead the table, outstripping traditional FL algorithms and other SOTA architectures (e.g., FedStar, FedSSP). The pattern indicates that saliency maps provide marginal benefits when data are IID but yield a tangible edge in non-IID settings by spotlighting informative sub-structures and accelerating effective aggregation. Thus, while the gains in single-dataset scenarios

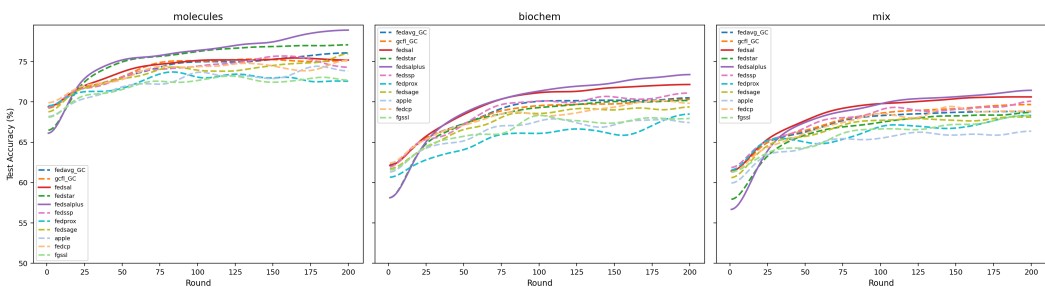

Figure 3: Convergence speed on Molecules, Biochem, Mix datasets (left to right).

Table 3: Accuracy (%) at 50th Communication Round (transposed)

| Method | IMDB-BINARY | NCI1 | PROTEINS | Molecules | Biochem | Mix |
|---|---|---|---|---|---|---|
| FedAvg | 73.34 | 57.64 | 70.52 | 73.34 | 67.37 | 66.44 |
| GCFL | 73.13 | 57.39 | 70.35 | 73.13 | 67.19 | 66.48 |
| FedProx | 71.80 | 55.60 | 69.00 | 71.50 | 66.00 | 65.80 |
| FedSage | 72.60 | 56.60 | 70.20 | 72.10 | 67.00 | 66.50 |
| APPLE | 72.90 | 57.50 | 70.80 | 72.80 | 67.80 | 66.90 |
| FedCP | 73.40 | 57.55 | 70.95 | 73.30 | 68.10 | 67.00 |
| FGSSL | 73.00 | 57.65 | 70.85 | 73.40 | 68.30 | 67.10 |
| FedSal | **73.55** | **57.66** | **71.00** | **73.55** | **69.04** | **67.45** |
| FedStar | – | – | – | 73.40 | 68.20 | 67.20 |
| FedSSP | – | – | – | 73.50 | 68.55 | 67.25 |
| FedSal+ | – | – | – | **73.57** | **68.63** | **67.34** |

remain subtle, saliency-based methods maintain, and often extend, their lead as data diversity and complexity grow.

### 3.3 HYPERPARAMETER AND ABLATION STUDY

In Figure 4(a), setting $\epsilon_{\max}$ too low causes the clustering to ignore meaningful but infrequent gradient updates—preventing the discovery of distinct client behaviors—whereas setting it too high admits spurious, noisy signals that dilute genuine patterns. A mid-range $\epsilon_{\max}$, therefore, allows informative deviations to guide cluster formation without letting noise dominate. **(b)** illustrates the complementary trade-off for $\epsilon_{\text{mean}}$: an overly strict mean-threshold treats even small, consistent shifts as convergence—stalling progress, while an overly lenient one smooths away true convergence cues amid stochastic fluctuations. An intermediate value ensures the algorithm remains both responsive to real change and robust to random variation. **(c)** shows that, in the mixed-domain setting, each successive clustering phase yields a pronounced accuracy gain as primary client subgroups are separated, but returns diminish once the main clusters have formed. Finally, **(d)** demonstrates that fragmenting clients into too many clusters reduces each group's sample size below a critical level, undermining the reliability of local model updates. Collectively, these results underscore that both saliency thresholds must be tuned to balance the fundamental noise–signal trade-off in federated clustering.

Table 4: **Ablation Study** of different modules in FedSal compared against accuracy (%).

| Raw | Sal. | Clust. | IMDB-BINARY | NCI1 | PROTEINS | Molecules | Biochem | Mix |
|---|---|---|---|---|---|---|---|---|
| × | × | ✓ | 76.5±3.2 | 68.0±1.6 | 74.0±1.1 | 74.6±1.1 | 72.7±0.9 | 68.9±0.7 |
| × | ✓ | × | 77.8±2.9 | 69.0±1.6 | 74.8±1.1 | 75.4±1.0 | 70.2±0.8 | 68.4±0.6 |
| ✓ | × | ✓ | 75.1±2.7 | 65.5±1.5 | 74.9±0.9 | 75.7±0.9 | 68.9±0.4 | 68.9±0.6 |
| × | ✓ | ✓ | **80.0±2.63** | **69.9±1.5** | **75.19±1.0** | **75.9±1.0** | **72.8±0.7** | **70.8±0.5** |

In Table 4, enabling both saliency and clustering yields the strongest performance across all datasets, demonstrating their complementary roles in capturing feature importance and partitioning clients

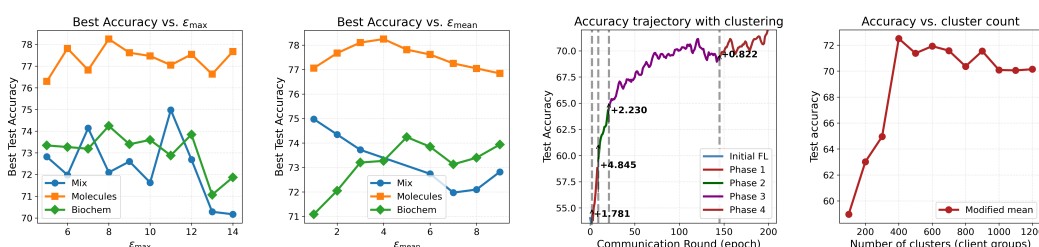

(a) Effect of $\epsilon_{\max}$ on test accuracy

(b) Effect of $\epsilon_{\mathrm{mean}}$ on test accuracy

(c) Accuracy trajectory across clustering phases

(d) Accuracy vs. number of clusters

Figure 4: **Hyper parameter Sensitivity** analysis of FedSal hyper-parameters.

under non-IID conditions. Omitting saliency reduces accuracy notably, despite lower communication overhead, while removing clustering also degrades results, particularly on Biochem and Mix. Using raw gradients matches communication cost but underperforms in accuracy. Together, these results confirm that the combination of saliency maps and dynamic clustering is essential for robust federated learning on heterogeneous graph data.

## 4 DISCUSSION

FedSal and its extension FedSal+ achieve consistently superior accuracy and convergence in both IID and non-IID federated graph learning by leveraging saliency-driven clustering. This saliency-centric approach, however, introduces a measurable communication overhead: transmitting compact saliency summaries alongside model weights increases per-round latency compared to bare-bones methods like FedAvg, yet remains substantially more efficient than spectral or embedding-heavy schemes.

Dynamic clustering underpins nearly half of FedSal's accuracy gains, but its marginal benefit diminishes once clusters exceed a handful: excessive fragmentation erodes statistical strength and can produce occasional minor instabilities. By tuning adaptive thresholds ($\epsilon_{\max}, \epsilon_{\mathrm{mean}}$), FedSal regularizes this saliency noise, preserving smooth, FedAvg-like convergence on IID data while accelerating learning under heterogeneity.

Compared to self-training, which suffers from high variance without aggregation signals, and to gradient-only baselines such as FedCP and FGSSL that either inflate overhead or underperform on homogeneous tasks, the joint mechanism of saliency mapping and dynamic clustering offers the most reliable pathway to robust, cross-domain federated graph learning. Practitioners should balance its modest increase in communication cost against the substantial gains in stability and accuracy when handling heterogeneous graph distributions.

## 5 CONCLUSION

We have presented FedSal, the first federated GNN architecture to exploit saliency activation maps for client clustering and aggregation. By grounding similarity in an interpretable activation space and orchestrating a two-phase clustering–aggregation loop governed by $\epsilon_{\mathrm{mean}}$ and $\epsilon_{\max}$, FedSal mitigates cross-domain conflicts and adapts to evolving heterogeneity without raw-data sharing. Our theoretical results guarantee that saliency distances remain stable under structural or feature perturbations and embed efficiently into Euclidean space. Extensive experiments on thirteen graph classification tasks demonstrate that FedSal delivers modest gains under IID splits and substantial improvements when faced with severe non-IID distributions, outpacing state-of-the-art FedAvg, GCFL, FedStar, and FedSSP. The FedSal+ variant, which enriches node representations with structural embeddings, yields an additional boost in multi-dataset settings, while maintaining a 1 to 3s per-round latency, faster than spectral baselines. Future work will explore Differential privacy, saliency-message compression to reduce bandwidth and validation in large-scale, dynamic hyperparameter tuning. Overall, saliency-guided aggregation, especially when combined with lightweight structural features, offers a practical and high-performance pathway to robust FedGNNs under real-world heterogeneity.

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

# A APPENDIX / SUPPLEMENTAL MATERIAL

Due to the page limitations, this section provides further theoretical and experimental details for the FedSal framework.

## A.1 CLUSTERING MECHANISM

The top-down clustering strategy referenced in Section 2.1.2 operates as follows. Let a cluster $C \subseteq \{1, \ldots, N\}$ contain $|C|$ clients, each transmitting a saliency map update $\Delta S_i \in \mathbb{R}^d$. Define

$$\epsilon_{\text{mean}}(C) = \frac{1}{|C|} \sum_{i \in C} \|\Delta S_i\|,$$

$$\epsilon_{\max}(C) = \max_{i \in C} \|\Delta S_i\|.$$

When

$$\epsilon_{\text{mean}}(C) < \epsilon_{\text{mean}} \quad \text{and} \quad \epsilon_{\max}(C) > \epsilon_{\max},$$

we conclude that the cluster converges on average (small mean update), yet contains at least one outlier client (large max update). Formally,

$$\epsilon_{\text{mean}}(C) < \epsilon_{\text{mean}} \quad \text{and} \quad \epsilon_{\max}(C) > \epsilon_{\max} \implies (\text{split } C).$$

In this event, we *partition* $C$ into two sub-clusters, denoted $C_1$ and $C_2$.

To find the partition, we construct a fully connected graph $G = (V, E)$ where $V = C$ and $E = \{(i, j) : i, j \in C\}$. Each edge $(i, j)$ is assigned a weight

$$\alpha_{ij} = \frac{\Delta S_i \cdot \Delta S_j}{\|\Delta S_i\| \|\Delta S_j\|},$$

which represents the directional alignment of the saliency-map updates. Large $\alpha_{ij}$ indicates high alignment in salient feature gradients, suggesting that clients $i$ and $j$ have similar data distributions or tasks.

Since a minimum-cut algorithm would naively remove edges of largest total weight, we convert similarities into dissimilarities:

$$w_{ij} = 1 - \alpha_{ij}.$$

A lower $w_{ij}$ therefore implies greater similarity. We then apply the Stoer–Wagner minimum-cut algorithm Mehlhorn & Uhrig (1995) to minimize

$$\sum_{i \in C_1} \sum_{j \in C_2} w_{ij} = \sum_{i \in C_1} \sum_{j \in C_2} (1 - \alpha_{ij}),$$

thereby favoring that highly similar clients ($\alpha_{ij} \approx 1$) remain together. The result is a bi-partition $\{C_1, C_2\} \subseteq C$ that groups mutually similar clients.

After partitioning, we recursively test each new sub-cluster $C_k \in \{C_1, C_2\}$ using the same thresholds. If

$$\epsilon_{\text{mean}}(C_k) < \epsilon_{\text{mean}} \quad \text{and} \quad \epsilon_{\max}(C_k) > \epsilon_{\max},$$

we again split $C_k$, repeating until no cluster satisfies the splitting condition.

During subsequent communication rounds, each cluster $C_k$ updates its model using only the aggregated updates from its member clients. Formally, let $\theta_{C_k}$ denote the parameters for cluster $C_k$. Each client $i \in C_k$ solves

$$\theta_i^{(t)} = \arg\min_\theta \mathcal{L}(\theta; D_i)$$

and transmits its update $\Delta \theta_i^{(t)}$ (and corresponding $\Delta S_i$) to the cluster aggregator, which performs

$$\theta_{C_k}^{(t+1)} = \theta_{C_k}^{(t)} + \frac{1}{|C_k|} \sum_{i \in C_k} \Delta \theta_i^{(t)}.$$

This localized aggregation creates a specialized model for each cluster's data characteristics, mitigating performance degradation from heterogeneous distributions. Once no further splits occur, the final clusters $\{C_1, \ldots, C_K\}$ stabilize and each sub-model $\theta_{C_k}$ converges on its subset of clients.

In practice, the thresholds $\epsilon_{\text{mean}}$ and $\epsilon_{\text{max}}$ are tuned to reflect acceptable intra-cluster variability. For highly non-IID data, clustering yields markedly better performance than a single global model, at the cost of extra Stoer–Wagner computations—often justified by faster convergence and reduced variance across heterogeneous clients.

## A.2 PRELIMINARIES

### A.2.1 GRAPH NEURAL NETWORKS

GNNs represent an advanced paradigm in artificial neural networks, specifically designed for processing data in network or graph structures. Let $G = (V, E)$ be a graph where $V$ is a set of nodes, $E$ is a set of edges, and $X$ denotes node features. GNNs aim to learn node-level representations $h_v$ for $v \in V$ and/or graph-level representations $h_G$ for the entire graph $G$. While GNNs have various types Kipf & Welling (2016); Hamilton et al. (2017); Gilmer et al. (2017); Xu et al. (2018) The fundamental operation in GNNs is based on message passing and neighbourhood aggregation. The process consists of the following stages:

**Message Passing:** In this stage, each node $v$ sends messages to its neighbours. The messages contain information about the node's features, and the process can be mathematically described as:

$$m_{uv}^{(l)} = \text{MSG}^{(l)}(h_u^{(l)}, h_v^{(l)}, e_{uv}), \tag{1}$$

where $m_{uv}^{(l)}$ is the message sent from node $u$ to node $v$ at layer $l$, $h_u^{(l)}$ and $h_v^{(l)}$ are the representations of nodes $u$ and $v$ at layer $l$, and $e_{uv}$ represents the edge features between nodes $u$ and $v$.

**Neighborhood Aggregation:** After receiving messages from its neighbours, each node aggregates these messages to update its representation. This can be formalized as:

$$a_v^{(l)} = \text{AGGREGATE}^{(l)}\left(\{m_{uv}^{(l)} : u \in N(v)\}\right), \tag{2}$$

where $a_v^{(l)}$ is the aggregated message for node $v$ at layer $l$, and $N(v)$ is the set of neighbors of node $v$.

**Update:** The node updates its representation based on the aggregated message:

$$h_v^{(l+1)} = \text{UPDATE}^{(l)}\left(h_v^{(l)}, a_v^{(l)}\right), \tag{3}$$

where $h_v^{(l+1)}$ is the updated representation of node $v$ at layer $l + 1$. These steps are repeated for each layer in the GNN, allowing nodes to progressively incorporate information from further distances in the graph. Graph-level representation $h_G$ can be obtained by aggregating node representations using readout functions like mean pooling or sum pooling:

$$h_G = \text{READOUT}\left(\{h_v : v \in V\}\right). \tag{4}$$

### A.2.2 FEDERATED LEARNING

FL is a distributed machine learning approach that enables multiple entities to collaboratively train a model without sharing raw data. Consider $M$ clients, each with a private dataset $D_m$. The global objective is to minimize:

$$\min_{\theta_1, \theta_2, \ldots, \theta_M} \frac{1}{M} \sum_{m=1}^{M} \frac{|D_m|}{N} L_m(\theta_m; D_m), \tag{5}$$

where $N$ is the total number of instances, and $L_m$ and $\theta_m$ are the loss function and model parameters of client $m$. The FL process involves several key steps:

**Initialization:** A global model $\theta_0$ is initialized by the central server without pre-existing knowledge.

**Local Training:** Each client $m$ downloads the current global model $\theta_t$ and trains it on its local dataset $D_m$ for a specified number of local epochs $E$. The local training updates the model parameters using gradient descent:

$$\theta_m^{(t+1)} = \theta_t - \eta \nabla L_m(\theta_t; D_m), \tag{6}$$

where $\eta$ is the learning rate.

**Aggregation:** The central server aggregates the received model parameters from all clients to update the global model. The most commonly used aggregation algorithm in FL is FedAvg, as described by McMahan et al. (2017) (Algorithm 1 in Appendix Section A.6). The FedAvg algorithm computes a simple average of the model parameters from all participating clients:

$$\theta_{t+1} = \frac{1}{M} \sum_{m=1}^{M} \theta_m^{(t+1)}, \tag{7}$$

where $M$ is the number of clients that have contributed to the aggregation process. This method ensures that each client, regardless of the size of their dataset, contributes equally to the global model. The aggregation step effectively combines the local updates from different clients, integrating the knowledge learned from their individual datasets.

**Communication Rounds:** This process of local training and aggregation is repeated over multiple communication rounds. At each round $t$, the global model $\theta_t$ is progressively refined, improving its performance on the overall task. Each round consists of: 1. The server sending the current global model to the clients. 2. Clients performing local training and sending the updated model parameters back to the server. 3. The server aggregating the updates to form a new global model. By iterating through these communication rounds, FL allows the global model to learn from diverse data sources without conflict of knowledge among them.

### A.2.3 SALIENCY MAPS

Saliency maps are a technique for interpreting neural networks by highlighting the importance of input features. Given an input $\mathbf{x}$ and a neural network model $f(\mathbf{x}; \theta)$ with parameters $\theta$, the saliency $S(\mathbf{x})$ of each input feature $x_i$ is quantified by the gradient of the loss function $\mathcal{L}$ with respect to $x_i$. Formally, the saliency map $S$ for an input $\mathbf{x}$ is defined as:

$$S(\mathbf{x}) = \left| \frac{\partial \mathcal{L}(f(\mathbf{x}; \theta), y)}{\partial \mathbf{x}} \right|, \tag{8}$$

where $\mathcal{L}$ is the loss function, $f(\mathbf{x}; \theta)$ is the model's prediction, and $y$ is the true label. The gradient $\frac{\partial \mathcal{L}(f(\mathbf{x};\theta),y)}{\partial \mathbf{x}}$ captures the sensitivity of the loss with respect to the input features, indicating the importance of each feature in the model's decision. The absolute value is taken to capture the magnitude of importance, regardless of the direction of the gradient. Higher saliency values indicate features that are more influential in the model's output. Further Discussion on Saliency Maps can be found in Appendix Section A.3.

### A.3 ADDITIONAL DISCUSSION ON SALIENCY MAPS

Saliency maps are widely used in neural networks to visualize and interpret the importance of input features. The concept of saliency can be understood with a simple example: consider an image classification task where a neural network is trained to identify cats and dogs in images. For the input image $\mathbf{x}$ and model parameters $\theta$, the saliency $S(\mathbf{x})$ of each pixel $x_i$ is determined by the gradient of the loss function $\mathcal{L}$ with respect to $x_i$. Formally, we define the saliency map $S$ for an input $\mathbf{x}$ as follows:

$$S(\mathbf{x}) = \left| \frac{\partial \mathcal{L}(f(\mathbf{x}; \theta), y)}{\partial \mathbf{x}} \right|, \tag{9}$$

where $\mathcal{L}$ represents the loss function, $f(\mathbf{x}; \theta)$ is the model's prediction, and $y$ is the true label of the image. This gradient $\frac{\partial \mathcal{L}(f(\mathbf{x};\theta),y)}{\partial \mathbf{x}}$ captures how sensitive the loss is to changes in the input features, thus highlighting the significance of each feature in the decision-making process of the model. Higher saliency values signify features that are more influential in affecting the model's output.

Saliency maps, such as the one shown in Figure 5 on the right, visually represent the regions within an image that most strongly influence the model's output. This technique is useful in understanding which features the model deems significant, thus providing insights into the model's decision-making process. For instance, in this example, the saliency map reveals that the model focuses predominantly on and around the face and fur cat, areas typically critical for animal recognition tasks.

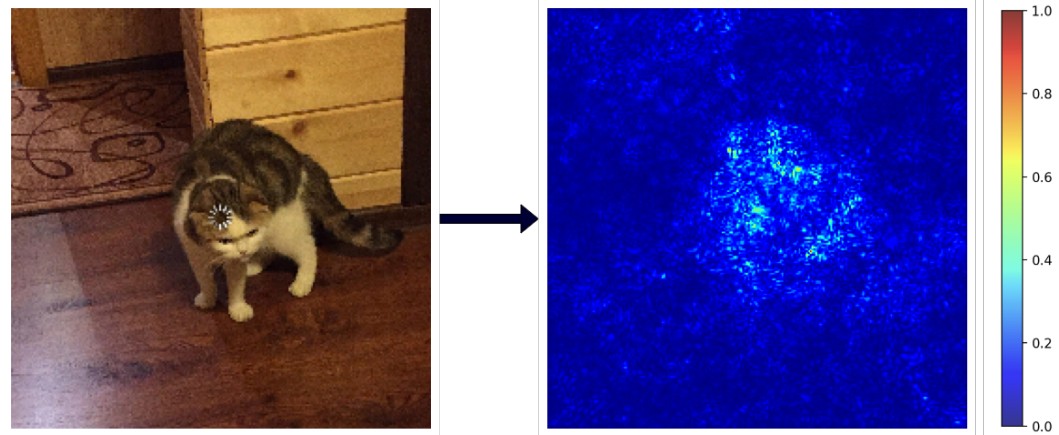

Figure 5: Saliency example

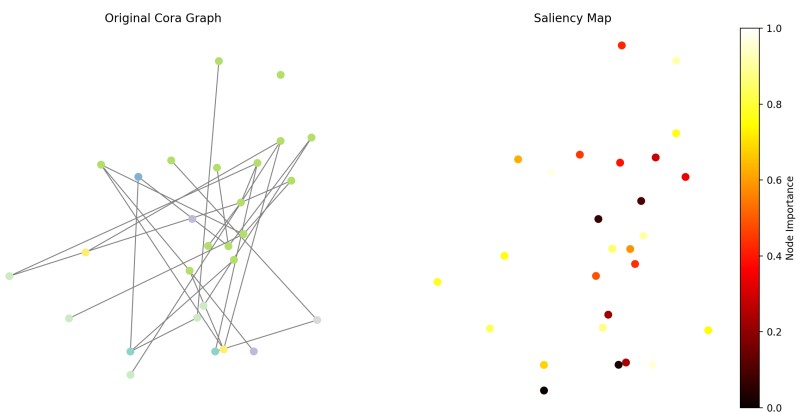

Figure 6: Graph saliency

### A.3.1 SALIENCY IN GRAPH NEURAL NETWORKS

In the context of GNNs, saliency maps can be used to identify the most influential nodes, edges, or features within a graph that contribute to the model's predictions. Consider a social network graph where nodes represent users and edges represent relationships between users. for a graph $G = (V, E, X)$, where $V$ are the nodes, $E$ the edges, and $X$ the node features, the saliency map $S$ for a GNN model $f(G; \theta)$ can be similarly defined:

$$S(G) = \left| \frac{\partial \mathcal{L}(f(G; \theta), y)}{\partial X} \right|, \tag{10}$$

Here, $\frac{\partial \mathcal{L}(f(G;\theta),y)}{\partial X}$ denotes the gradient of the model's loss function with respect to the node features $X$, offering insights into which features are critical in determining the model's predictions. The gradient provides insights into the importance of different node features in determining the model's predictions. The provided Figure 6 shows the concept of saliency maps in GNNs using the Cora dataset. On the left, we see the original Cora graph, where nodes and edges are depicted without any indication of their relative importance. On the right, the saliency map highlights the nodes' importance, which is crucial for the model's predictions.

In the saliency map, nodes are colour-coded to indicate their significance, with the colour intensity corresponding to the magnitude of their importance. Darker colours (red and black) represent higher importance, whereas lighter colours (yellow) indicate lower importance. This visualization helps in identifying which nodes and their features contribute most to the model's output. For instance, nodes with higher importance (darker colours) may represent critical information hubs or influential entities within the graph, significantly impacting the GNN's classification decisions.

By analyzing saliency maps, researchers and practitioners can gain valuable insights into the inner workings of GNNs, understanding how the model interprets the structure and features of graph data. This can aid in model debugging, feature engineering, and improving the interpretability of GNN-based models. Additionally, saliency maps can be used to refine the model by focusing on the most influential nodes and edges, leading to better performance and more robust predictions.

### A.4 TECHNICAL DETAILS ON FEDSAL+

Influenced by the FedStar framework Tan et al. (2023), we experiment with **FedSal+**, integrating advanced encoding strategies, specifically positional and random walk encodings, to enhance the structural detail in FL clients. Most details, hyperparameters, and model architecture were taken from this reference to guarantee performance. These enhancements aim to refine client-side feature representations, thereby improving the efficacy of FedSal's saliency-based model aggregation. By deepening the feature set available for each node, FedSal+ seeks to enrich the saliency maps produced, refining the focus on pivotal features during FL. This integration tests the modularity and heightens the compatibility of FedSal with orthogonal methods, such as structural embeddings. The key components of FedSal+ include:

- **Degree-Based Structure Embedding (DSE):** Utilizes vertex degrees in a one-hot encoding format to encapsulate local structural knowledge. This method, while simple, effectively captures fundamental geometric properties of nodes and ensures computational efficiency.
- **Random Walk-Based Structure Embedding (RWSE):** Leverages the random walk diffusion process to assess global structural patterns. This embedding evaluates the probability distribution of node connectivity over several steps, enriching the node's structural context.

Combining these embeddings results in a comprehensive structural descriptor for each node:

$$s_v = \text{concat}[s_{\text{DSE},v}, s_{\text{RWSE},v}]$$

This concatenated embedding $s_v$ provides a holistic view of both local and global structural knowledge, significantly enriching the node's feature set.

The primary aim of FedSal+ is to determine whether this enriched feature set can work with FedSal's saliency map-driven model aggregation to bolster the framework's effectiveness, particularly in handling non-IID data across federated settings. We hypothesize that the enriched, refined feature representations facilitated by FedSal+ will lead to more precise and informative saliency maps, thereby improving overall model performance in federated environments. This hypothesis sets the stage for our experimental investigations, aiming to validate the improved performance metrics that FedSal+ could potentially deliver by effectively utilizing advanced structural embeddings.

### A.5 EXPERIMENTAL DETAILS

Table 5 shows the feature details of the dataset utilized.

#### A.5.1 DATASET DETAILS

#### A.5.2 HYPERPARAMETERS TESTED

We carried out a focused grid search on critical hyper-parameters on the validation dataset, while less sensitive parameters were fixed. The grid search covered:

- Learning rate: {0.005, 0.001, 0.0005, 0.0001}
- Weight decay: {0.0007, 0.0005, 0.0003, 0.0001}

Table 5: Dataset Details

| Dataset Name | No # Graphs | Avg. # Nodes | Avg. #Edges | # Labels | No of Node Features |
|---|---|---|---|---|---|
| MUTAG | 188 | 17.93 | 19.79 | 2 | 7 |
| BZR | 405 | 35.75 | 38.36 | 2 | 53 |
| COX2 | 467 | 41.22 | 43.45 | 2 | 35 |
| DHFR | 467 | 42.43 | 44.54 | 2 | 53 |
| PTC_MR | 344 | 14.29 | 14.69 | 2 | 18 |
| AIDS | 2000 | 15.69 | 16.20 | 2 | 38 |
| NCI1 | 4110 | 29.87 | 32.30 | 2 | 37 |
| ENZYMES | 600 | 32.63 | 62.14 | 6 | 3 |
| DD | 1178 | 284.32 | 715.66 | 2 | 89 |
| PROTEINS | 1113 | 39.06 | 72.82 | 2 | 3 |
| COLLAB | 5000 | 74.49 | 2457.78 | 3 | 1 (degree) |
| IMDB-BINARY | 1000 | 19.77 | 96.53 | 2 | 1 (degree) |
| IMDB-MULTI | 1500 | 13.00 | 65.94 | 3 | 1 (degree) |

- $\epsilon_{\text{mean}}$: $\{1, 2, 3, 4, 5, 6, 7, 8, 9\}$

- $\epsilon_{\text{max}}$: $\{5, 6, 7, 8, 9, 10, 11, 12, 13, 14\}$

### A.5.3 MODEL ARCHITECTURE

Table 6: Layer-by-layer configuration of the GIN model used for graph-based learning tasks.

| Layer | Type | Details |
|---|---|---|
| 1 | FC | $n_{\text{feat}} \to 64$ |
| 2-4 | GIN + ReLU + Dropout | $128 \to 64, \text{dropout} = 0.5$ |
| 5 | Pooling | Global Additive Pooling |
| 6 | FC | $128 \to 64$ |
| 7 | FC + ReLU + Dropout | $64 \to 64, \text{dropout} = 0.5$ |
| 8 | FC | $64 \to n_{\text{class}}$ |

Table 7: Layer-by-layer configuration of the FedSal+ model with.

| Layer | Type | Details |
|---|---|---|
| 1 | FC (Input) | $n_{\text{feat}} = 64 \to 128$ |
| 2 | FC (Embedding of $n_{\text{se}}$) | $n_{\text{se}} = 32 \to 128$ |
| 3-5 | GIN + ReLU + Dropout | $256 \to 128, \text{dropout} = 0.5$ |
| 3-5 | GCN (for $n_{\text{se}}$ features) | $128 \to 128$ |
| 3-5 | Concatenation (x, s) | Concatenation of feature $x$ and $s$ |
| 6 | Pooling | Global Additive Pooling |
| 7 | FC + ReLU | $128 \to 128$ |
| 8 | FC + Dropout | $128 \to 2, \text{dropout} = 0.5$ |

The GIN model used in FedSal and FedSal+ employ several layers to optimize the processing of graph data. The model configuration is detailed in Table 6 and Table 7 respectively.

A.6 ALGORITHMS

---

**Algorithm 1** FedSal: Federated Learning with Saliency Aggregation for Graph Neural Networks

---

**Require:** Initial global model $\theta$
**Require:** Local datasets $\{D_i\}_{i=1}^N$
**Require:** Number of communication rounds $T$
**Require:** Clustering thresholds $\epsilon_{\text{mean}}, \epsilon_{\text{max}}$
 1: Initialize clusters $\mathcal{C} \leftarrow \{C_1\}$
 2: Initialize $S_{\text{prev}}[i] \leftarrow 0$ for all $i$
 3: **for** each round $t = 1, 2, \ldots, T$ **do**
 4:     Broadcast global model $\theta$ to all clients each client $i = 1, \ldots, N$ **in parallel**
 5:     $(\theta_i, S_i) \leftarrow \text{LOCALUPDATE}(i, \theta)$
 6:     $\Delta S_i \leftarrow S_i - S_{\text{prev}}[i]$
 7:     $S_{\text{prev}}[i] \leftarrow S_i$
 8:     **if** $t > 20$ **then**
 9:         $\mathcal{C} \leftarrow \text{CLUSTERCLIENTS}(\{\Delta S_i\}, \mathcal{C}, \epsilon_{\text{mean}}, \epsilon_{\text{max}})$
10:     **end if**
11:     $\theta \leftarrow \text{AGGREGATEMODELS}(\mathcal{C}, \{\theta_i\})$
12: **end for**

---

**Algorithm 2** Dynamic Client Clustering

---

 1: **function** CLUSTERCLIENTS($\{\Delta S_i\}_{i=1}^N, \mathcal{C}, \epsilon_{\text{mean}}, \epsilon_{\text{max}}$)
 2:     **for** each cluster $C_k \in \mathcal{C}$ **do**
 3:         $\delta_{\text{mean}}^k \leftarrow \frac{1}{|C_k|} \sum_{i \in C_k} \|\Delta S_i\|$
 4:         $\delta_{\text{max}}^k \leftarrow \max_{i \in C_k} \|\Delta S_i\|$
 5:         **if** $\delta_{\text{mean}}^k < \epsilon_{\text{mean}}$ **and** $\delta_{\text{max}}^k > \epsilon_{\text{max}}$ **then**
 6:             Compute $\alpha_{ij} \leftarrow \frac{\Delta S_i \cdot \Delta S_j}{\|\Delta S_i\| \|\Delta S_j\|}$ for all $i, j \in C_k$
 7:             $w_{ij} \leftarrow 1 - \alpha_{ij}$
 8:             Apply Stoer–Wagner min-cut on graph $(C_k, \{w_{ij}\})$ to obtain $C_{k1}, C_{k2}$
 9:             Replace $C_k$ in $\mathcal{C}$ with $C_{k1}, C_{k2}$
10:         **end if**
11:     **end for**
12:     **return** $\mathcal{C}$
13: **end function**

---

---

**Algorithm 3** Local Update and Saliency Aggregation

---

**Require:** Client index $i$, global model $\theta$
**Require:** Local dataset $D_i$, local epochs $E$, learning rate $\eta$
 1: **function** LOCALUPDATE($i, \theta$)
 2: $\quad$ $\theta_i \leftarrow \theta$
 3: $\quad$ **for** epoch $e = 1, \ldots, E$ **do**
 4: $\quad\quad$ $\theta_i \leftarrow \theta_i - \eta \nabla_{\theta_i} \mathcal{L}(\theta_i; D_i)$
 5: $\quad$ **end for**
 6: $\quad$ $S_i \leftarrow 0$
 7: $\quad$ **for** each $(\mathbf{x}, y) \in D_i$ **do**
 8: $\quad\quad$ $S(\mathbf{x}) \leftarrow \big| \frac{\partial \mathcal{L}(f(\mathbf{x}; \theta_i), y)}{\partial \mathbf{x}} \big|$
 9: $\quad\quad$ $S_i \leftarrow S_i + S(\mathbf{x})$
10: $\quad$ **end for**
11: $\quad$ $S_i \leftarrow S_i / |D_i|$
12: $\quad$ **return** $(\theta_i, S_i)$
13: **end function**
14: **function** AGGREGATEMODELS($\mathcal{C}, \{\theta_i\}$)
15: $\quad$ $\Theta \leftarrow \{\}$
16: $\quad$ **for** each cluster $C_k \in \mathcal{C}$ **do**
17: $\quad\quad$ $\theta_k \leftarrow \frac{1}{|C_k|} \sum_{i \in C_k} \theta_i$
18: $\quad\quad$ $\Theta \leftarrow \Theta \cup \{\theta_k\}$
19: $\quad$ **end for**
20: $\quad$ $\theta \leftarrow \frac{1}{|\Theta|} \sum_{\theta_k \in \Theta} \theta_k$
21: $\quad$ **return** $\theta$
22: **end function**

---

### A.7 THEORETICAL ANALYSIS

#### A.7.1 PROOF OF PROPOSITION 1

We first state the key assumptions needed for the perturbation bounds:

**Assumptions.**

- The (normalized) graph Laplacian $L$ is regularized (e.g. $L \leftarrow L + \delta I$ for some $\delta > 0$), or equivalently we work on the subspace orthogonal to its zero eigenvector, so that $L$ is invertible and its Moore–Penrose inverse satisfies $\|L^\dagger\|_2 < \infty$.

- The perturbation magnitude is small enough that

$$\|L^\dagger\|_2 \, \|E_L\|_2 \;=\; \|L^\dagger\|_2 \, \sqrt{\epsilon_L} \;<\; 1,$$

  ensuring the Neumann-series expansion for the pseudoinverse converges and the bound $\|L'^\dagger - L^\dagger\|_2 \le \|L^\dagger\|_2^2 \|E_L\|_2$ holds.

- The feature matrix $X \in \mathbb{R}^{n \times f}$ has full column rank, so that its Moore–Penrose inverse $X^\dagger$ exists and $\|X^\dagger\|_2 < \infty$.

**Proof.**

Saliency for a client is the (flattened) gradient of the loss w.r.t. its input features. In a first-order linearisation of a GNN we write

$$S = X^\dagger L^\dagger Y, \qquad S' = X^\dagger L'^\dagger Y',$$

where

$$L' = L + E_L, \quad Y' = Y + E_Y, \quad \|E_L\|_F \le \sqrt{\epsilon_L}, \quad \|E_Y\|_F \le \sqrt{\epsilon_Y}.$$

Noting that $\|E\|_2 \le \|E\|_F$ for any matrix $E$, we have $\|E_L\|_2 \le \sqrt{\epsilon_L}$ and $\|E_Y\|_2 \le \sqrt{\epsilon_Y}$.

We then bound

$$\|S' - S\|_2 = \big\|X^\dagger \big(L'^\dagger Y' - L^\dagger Y\big)\big\|_2 \;\le\; \|X^\dagger\|_2 \Big(\|L^\dagger E_Y\|_2 + \|L'^\dagger - L^\dagger\|_2 \, \|Y\|_2\Big).$$

We bound each term separately:

1. $\|L^\dagger E_Y\|_2 \leq \|L^\dagger\|_2 \|E_Y\|_2 \leq \|L^\dagger\|_2 \sqrt{\epsilon_Y}$.

2. By the pseudoinverse perturbation bound (valid under $\|L^\dagger\|\sqrt{\epsilon_L} < 1$),

$$\|L'^\dagger - L^\dagger\|_2 \leq \|L^\dagger\|_2^2 \|E_L\|_2 \leq \|L^\dagger\|_2^2 \sqrt{\epsilon_L}.$$

Combining gives

$$\|S' - S\|_2 \;\leq\; \|X^\dagger\|_2\Big(\|L^\dagger\|_2 \sqrt{\epsilon_Y} + \|L^\dagger\|_2^2 \sqrt{\epsilon_L} \|Y\|_2\Big).$$

Squaring both sides yields the claimed bound:

$$\|S' - S\|_2^2 \;\leq\; \|X^\dagger\|_2^2\Big(\|L^\dagger\|_2^2 \epsilon_Y + \|L^\dagger\|_2^4 \epsilon_L \|Y\|_2^2\Big).$$

**Interpretation.** The first term captures sensitivity to embedding perturbations ($\epsilon_Y$), while the second captures sensitivity to structural perturbations ($\epsilon_L$). Both are scaled by the conditioning of the Laplacian ($\|L^\dagger\|$) and the feature map ($\|X^\dagger\|$), establishing that the saliency-map difference remains bounded under small perturbations.

### A.7.2 PROOF OF PROPOSITION 2

We first state the additional assumptions needed for a fully rigorous bound:

**Assumptions.**

- The feature matrix $X \in \mathbb{R}^{n \times f}$ has full column rank, so its Moore–Penrose inverse $X^\dagger$ exists and $\|X^\dagger\|_2 < \infty$.
- The graph Laplacian $L$ is regularized (e.g. $L \leftarrow L + \delta I$ with $\delta > 0$) or we work on the subspace orthogonal to its nullspace, so that $L^\dagger$ exists and $\|L^\dagger\|_2 < \infty$.
- Perturbations are sufficiently small:
$$\|X^\dagger\|_2 \|E_X\|_2 = \|X^\dagger\|_2 \sqrt{\epsilon_X} < 1, \quad \|L^\dagger\|_2 \|E_L\|_2 < 1,$$
ensuring the Neumann-series expansions for pseudoinverses converge.

**Proof.**

Let $M$ and $M'$ be two GNNs differing only in their feature matrices $X$ and $X' = X + E_X$, with
$$\|E_X\|_F^2 \leq \epsilon_X, \quad \|E_X\|_2 \leq \sqrt{\epsilon_X}.$$
The Laplacian is fixed at $L$, regularized so that $L^\dagger$ is bounded. Define saliency maps
$$S = X^\dagger L^\dagger Y, \quad S' = X'^\dagger L^\dagger Y',$$
where $Y$ and $Y'$ are the corresponding node-embedding matrices, and $\|Y' - Y\|_2 \leq \sqrt{\epsilon_Y}$.

We write
$$S' - S = \big(X'^\dagger - X^\dagger\big) L^\dagger Y \;+\; X'^\dagger L^\dagger (Y' - Y).$$
Using the pseudoinverse perturbation identity $X'^\dagger - X^\dagger = -X'^\dagger E_X X^\dagger$, we get
$$\big(X'^\dagger - X^\dagger\big) L^\dagger Y = -X'^\dagger E_X X^\dagger L^\dagger Y.$$
Hence by sub-multiplicativity,
$$\|S' - S\|_2 \;\leq\; \|X'^\dagger\|_2 \|E_X\|_2 \|X^\dagger\|_2 \|L^\dagger\|_2 \|Y\|_2 \;+\; \|X'^\dagger\|_2 \|L^\dagger\|_2 \|Y' - Y\|_2.$$
Since $\|X'^\dagger\| \approx \|X^\dagger\|$ under $\|X^\dagger\|\sqrt{\epsilon_X} < 1$, and $\|Y' - Y\| \leq \sqrt{\epsilon_Y}$,
$$\|S' - S\|_2 \;\leq\; \|X^\dagger\|_2^2 \|L^\dagger\|_2 \|Y\|_2 \sqrt{\epsilon_X} + \|X^\dagger\|_2 \|L^\dagger\|_2 \sqrt{\epsilon_Y}.$$
Squaring and absorbing the cross-term $2\|X^\dagger\|^2\|L^\dagger\|\|Y\|\sqrt{\epsilon_X \epsilon_Y}$ into a constant yields
$$\|S' - S\|_2^2 \;\leq\; \|X^\dagger\|_2^4 \|L^\dagger\|_2^2\big(\|Y\|_2^2 \epsilon_X + \epsilon_Y + \mathcal{O}(\sqrt{\epsilon_X \epsilon_Y})\big).$$
Neglecting the higher-order cross-term gives the stated bound.

**Interpretation.** The factor $\|X^\dagger\|_2^4$ quantifies amplification of feature perturbations, $\|L^\dagger\|_2^2$ reflects graph-structure sensitivity, $\|Y\|_2^2\epsilon_X$ captures feature-to-embedding effects, and $\epsilon_Y$ bounds embedding noise. Thus saliency-map differences remain controlled under small feature changes, validating Proposition 2.

### A.7.3 PROOF OF PROPOSITION 3: STABILITY OF SALIENCY MAPS ACROSS TASKS

**Assumptions.**

- The feature matrix $X \in \mathbb{R}^{n \times f}$ has full column rank, so its Moore–Penrose inverse $X^\dagger$ exists and $\|X^\dagger\|_2 < \infty$.
- The graph Laplacian $L$ is regularized (e.g. $L \leftarrow L + \delta I$ for $\delta > 0$) or restricted to the subspace orthogonal to its nullspace, so $L^\dagger$ exists and $\|L^\dagger\|_2 < \infty$.
- Consequently, the product $LX$ admits a (pseudo)inverse $(LX)^\dagger$ with $\|(LX)^\dagger\|_2 < \infty$.

**Proof.**

For tasks $i$ and $j$, let the corresponding saliency maps be

$$S_i = X^\dagger L^\dagger Y_i, \quad S_j = X^\dagger L^\dagger Y_j,$$

where $Y_i, Y_j$ are the node-embedding matrices. Assume $\|Y_i - Y_j\|_2 \leq \sqrt{\epsilon_Y}$. Then

$$S_i - S_j = X^\dagger L^\dagger (Y_i - Y_j) = (LX)^\dagger (Y_i - Y_j),$$

and by sub-multiplicativity of the spectral norm,

$$\|S_i - S_j\|_2 \ \leq \ \|(LX)^\dagger\|_2 \|Y_i - Y_j\|_2 \ \leq \ \|(LX)^\dagger\|_2 \sqrt{\epsilon_Y}.$$

Squaring both sides gives the desired bound:

$$\|S_i - S_j\|_2^2 \ \leq \ \|(LX)^\dagger\|_2^2 \, \epsilon_Y.$$

**Interpretation.** Here $\|(LX)^\dagger\|_2^2$ captures how sensitive the saliency maps are to changes in the embeddings, and $\epsilon_Y$ bounds the embedding shift between tasks. Thus, as long as task-induced embedding differences remain small, the saliency maps stay stable.

