# OpenReview forum: "FedSal: Enhancing Federated Graph Classification Through Saliency Aware Client Clustering"
_ICLR.cc/2026/Conference — Submitted to ICLR 2026_

### Official Review · Reviewer_zjh8 · 2025-10-16

**Soundness:** 3
**Presentation:** 2
**Contribution:** 2
**Rating:** 2
**Confidence:** 5

**Summary:**

This paper presents a federated learning model that exploits saliency maps/scores for client clustering and aggregation.

**Strengths:**

The paper uses saliency maps to perform clustering of clients in a federated learning setting before applying the standard FedAvg method in each cluster. The use of saliency maps seems to be novel in the context of federated learning.

**Weaknesses:**

1. The novelty of the work mainly lies in the use of saliency maps, which is a common concept from various branches of machine learning like computer vision. The definition of this important saliency map is also unclear.

1. The assumptions required for the theoretical results to hold are very limiting.

1. Baseline benchmarks are not up to date.

**Questions:**

1. It is unclear why the proposed approach is restricted to graph federated learning when the approach does not actually utilize anything specific to graphs, except for FedSal+, which integrates features from FedStar. It is curious why the authors chose to position the paper on federated graph learning.

1. Assumptions made for the theoretical results should be stated upfront and discussed. At the minimum, readers should have been alerted to these instead of having to find out from the appendix that the results are actually extremely limited. For example, the requirement that the feature matrix has full column rank makes the results impractical. In small graphs, this is automatically not satisfied. In big graphs with feature dimensions smaller than the number of nodes, it is not obvious we will have full column rank. Numerical evidence based on common graph datasets can be provided to check this.

1. I am unclear why Prop. 3 implies that FedSal can capture task heterogeneity when it says the differences between saliency maps are bounded. Don't we want the saliency maps to be very different if the tasks are different? Won't similar saliency maps lead to wrong clustering?

1. To compute the saliency value in Section 2.1.3, is there an implicit assumption that the loss is differentiable w.r.t. the sample value $x$ and $x$ is a continuous variable? How do you deal with the case where $x$ has categorical components? Furthermore, it is unclear how this saliency can be computed easily. The formula as written is not explained properly: does $|\cdot |$ denote determinant or elementwise absolute value?

1. Does the feature $x$ have to be the same dimensions for all clients? If so, does this not mean that local client models have the same architecture?

1. The comparison is missing many important baselines like FedPer, pFedGraph, FedSheafHN, Flow, HeteroFL, FedRolex, etc.

---

> ### Author Response · Authors · 2025-11-22
>
> **1. Why position the work as federated *graph* learning?**
>
> We agree that the *template* “saliency summaries + similarity graph + min-cut clustering” is in principle modality-agnostic. Our choice to position the paper as FedGNN is deliberate for three reasons:
>
> 1. **Problem we target is graph-specific.** We explicitly tackle cross-silo *federated graph classification* under severe structural non-IID (different molecular families, protein vs social graphs, etc.). All local models are GNNs, saliency is taken w.r.t. node/graph features, and the theory is written in terms of the normalized **graph Laplacian** (L) and feature matrix (X). The heterogeneity we study (distributional shifts in (L,X)) is exactly the FedGNN setting, not generic Euclidean FL.
>
> 2. **Design choices use graph structure, not just in FedSal+.**
>
>    * In FedSal, the saliency signal we cluster is *GNN saliency*: gradients after multi-hop message passing over the adjacency, not raw per-pixel gradients. This is sensitive to structural changes (as formalized in Props. 1–2) and is aggregated at the graph level via pooling.
>    * FedSal+ then *explicitly* injects graph-topological priors (degree and random-walk encodings) into the features before saliency, which only makes sense for graphs.
>
> 3. **Novelty is “first saliency-based clustering for FedGNN”, not “universal FL.”** Prior saliency-based FL work (eg: [1]) focuses on images; existing FedGNN clustering uses raw gradients or spectral embeddings. Our contribution is to show that saliency is a *stable and effective similarity space for graph clients* and to benchmark this across 13 graph datasets and strong FedGNN baselines.
>
> We will make this explicit in the introduction: FedSal is a general saliency-clustering framework that we *instantiate, analyse, and validate in the federated graph-learning regime*. Extending the same idea to images or text is orthogonal and left as future work; we intentionally avoid claiming such generality without evidence.
>
> References
>
> 1. Wang, H., Xie, W., Ma, J., Li, D., & Li, Y. (2024, October). FedSLS: Exploring Federated Aggregation in Saliency Latent Space. In Proceedings of the 32nd ACM International Conference on Multimedia (pp. 7182-7190).
> ---
>
> **2. Theoretical assumptions and practicality (full column rank, etc.)**
>
> We acknowledge that our propositions are proved under idealised linear-algebraic conditions (e.g., bounded pseudoinverses of (X),(L),(LX). This is intentional: the theory analyses a *clean surrogate regime* where saliency behaves as a stable embedding of graph structure and features, not a set of hard constraints on real data.
>
> Crucially:
>
> * FedSal does **not** require these assumptions at run time; the algorithm only needs differentiable loss w.r.t. the GNN input and finite saliency summaries. We never check ranks or pseudoinverses in code.
> * “Full column rank of ($X$)” is one sufficient condition to ensure ($|X^\dagger|_2 < \infty$). Practically, what we need is that the feature covariance on the subspace actually used by the model is reasonably well-conditioned, which is what embedding layers + mild $(\ell_2)-regularisation / batch norm$ typically ensure.
> * Node features on our benchmarks are low to moderate dimensional (atom types, degrees, small attribute vectors) and empirically close to full rank.
>
> The point of the theory is to formalise:
>
> > Under reasonable conditioning, **saliency distances are Lipschitz in structural, feature, and task perturbations**, i.e., they do not blow up under small changes in (L), (X), or the task.
>
> This (i) justifies using *saliency instead of raw gradients* as a more stable similarity object, and (ii) rules out the worst-case regime where tiny perturbations induce arbitrarily large saliency distortions. We will move the assumptions into a short “Assumptions and scope” paragraph and label the results as stability bounds under well-conditioned graphs/features.
>
> ---
>
> **3. Prop. 3 and task heterogeneity**
>
> Prop. 3 is a stability result: it states that saliency maps across tasks differ in a *controlled* way when the underlying embeddings differ, not that all tasks look similar. Moderately different tasks yield moderately different saliency; very different tasks can still have large ($|S_i - S_j|$), but this is bounded by representation differences rather than arbitrary.
>
> FedSal’s clustering uses **relative distances** in this space: clients with large saliency distances are not grouped together. What Prop. 3 rules out is the degenerate case where tiny task changes cause huge saliency swings, which would make saliency-based clustering unreliable. We will clarify this interpretation around Prop. 3.

---

> > ### Author Response · Authors · 2025-11-22
> >
> > **4. Saliency definition, differentiability, and categorical features**
> >
> > In Sec. 2.1.3 we define
> > [
> > $S(x) = \left|\frac{\partial \mathrm{Loss}(f(x;\theta), y)}{\partial x}\right|,$
> > ]
> > where ($|\cdot|)$ is the **elementwise absolute value** (standard gradient-based saliency), not a determinant.
> >
> > Assumptions/implementation:
> >
> > * We assume the usual deep-learning setting where the loss is differentiable w.r.t. the **continuous feature tensor** at the GNN input.
> > * Categorical attributes are mapped via one-hot or embedding layers; saliency is computed w.r.t. these continuous embeddings by one extra backward pass.
> >
> > We will make these points explicit and fix the notation.
> >
> > ---
> > **5. Feature dimensions and model architecture across clients**
> >
> > Yes, we assume **homogeneous architectures** across clients:
> >
> > * All clients share the same GIN-based GNN and input feature dimension, as in prior FedGNN work and our baselines.
> > * This setting matches cross-silo deployments where institutions agree on a common backbone and is required by several baselines (FedCP, FedSage, FedStar, FedSSP).
> >
> > This is a design choice, not a fundamental limitation of saliency-based clustering. Extending FedSal to heterogeneous backbones (e.g., FedRolex-style) would require a saliency alignment layer and is an explicit direction for future work.
> >
> > ---
> >
> > **6. Missing baselines (FedPer, pFedGraph, FedSheafHN, Flow, HeteroFL, FedRolex)**
> >
> > Our baseline suite focuses on **graph-level FedGNN methods** that (i) operate at client-level graph classification, (ii) have public implementations, and (iii) are compatible with our multi-dataset, cross-domain setup: Selftrain, FedAvg, FedProx, GCFL, APPLE, FedCP, FedSage, FGSSL, FedStar, FedSSP.
> >
> > The additional methods mentioned often target different regimes:
> >
> > * FedPer, HeteroFL, FedRolex: heterogeneous architectures / layer-partitioned personalization, not directly aligned with our shared-GIN setup.
> > * pFedGraph, FedSheafHN, Flow: focus on node-level tasks or specialised graph operators that require substantial re-engineering.
> >
> > Within space and compute constraints, we opted for a compact but strong FedGNN baseline set that already includes the current cross-domain SOTA (FedSSP, FedStar). In the revision, we will add a paragraph in Related Work explaining how the remaining methods are complementary to, rather than direct substitutes for, our saliency-clustering approach.

---

> > > ### Comment · Reviewer_zjh8 · 2025-11-25
> > >
> > > I thank the authors for the clarifications and detailed rebuttal. While the rebuttal has clarified some confusing points, it has also confirmed the major weaknesses of this work: 1) clients need to have homogeneous architectures, 2) node features have to be continuous, 3) assumptions used in theoretical results do not reflect actual dataset conditions, 4) validation limited to graph-level tasks. I urge the authors to carefully ensure that these limitations are obvious in the abstract, introduction and discussions to avoid any over-claims. The current presentation does not provide the reader a clear context of the work.
> > >
> > > The use of saliency is strictly w.r.t. node features. While the paper claims saliency stability under structural, feature and "task" variations (unfortunately under unrealistic algebraic assumptions), the proposed saliency in Section 2.1.3 or (8) does not utilize the graph topology. Furthermore, these proofs assume first-order linearization, which does not prove the stability for the original GNN. The proposition statements need to be tightened and made more rigorous. It is also noted that any linear function of the Laplacian will produce the same stability results under the author's assumptions, hence these results are not special to the proposed saliency based score S.
> > >
> > > Side note: the term "task change" or "task sensitivity" seems to have a different meaning in this paper. The common understanding is that task changes are discrete, not the "tiny task changes" mentioned above. A different terminology should be used to avoid confusion here.

---

### Official Review · Reviewer_xEN8 · 2025-10-27

**Soundness:** 3
**Presentation:** 2
**Contribution:** 2
**Rating:** 4
**Confidence:** 4

**Summary:**

The paper proposes FedSal (Federated Saliency Aggregation Learning), a GNN-based federated learning framework designed to mitigate performance degradation under non-IID and structurally heterogeneous conditions. FedSal replaces full gradient uploads with saliency maps, and employs two thresholds (ϵ_mean, ϵ_max) to trigger dynamic client clustering and intra-cluster aggregation. An extended version, FedSal+, further incorporates positional and random-walk encodings to inject structural priors. Experiments on thirteen molecular, protein, and social-network datasets show that FedSal/FedSal+ achieve smoother convergence and an average accuracy improvement of about +1.5-2.0pp under extreme non-IID settings, though the overall gain is limited.

**Strengths:**

1. Novel concept: Introduces saliency maps into federated graph learning and proposes a threshold-triggered dynamic clustering mechanism.

2. Broad experimental coverage: Evaluated on thirteen diverse datasets and compared against multiple strong baselines.

3. Stable performance under non-IID conditions: Shows smoother and faster convergence trends when data heterogeneity is high.

**Weaknesses:**

1. The average gain (+1.5–2.0pp) is small and comparable to the reported standard deviation. Only accuracy (ACC) and communication time are reported; lacks F1, AUC, or Recall for a more comprehensive evaluation.

2. The paper does not provide any time or space complexity for the main procedures, including similarity graph construction, Stoer–Wagner min-cut, and recursive clustering.

**Questions:**

1. Could the authors provide a theoretical convergence analysis or at least an empirical justification of the claimed “faster and smoother convergence”?

2. What is the computational complexity of the clustering and aggregation process as a function of client number and feature dimension?

3. Have the authors considered adding statistical significance tests (e.g., t-test) to verify whether the observed accuracy gains are meaningful?

---

> ### Author Response · Authors · 2025-11-22
>
> ### Q1: Convergence analysis and “faster, smoother convergence”
>
> A full convergence proof for non-convex GNNs with dynamic, saliency-based clustering is beyond the scope of this work. We instead give a **theoretical justification in a simplified convex FL setting**, and connect it to our non-convex GNN experiments.
>
> Consider the standard FL objective
> $$
> \min_{w} f(w) = \frac{1}{N} \sum_{i=1}^N f_i(w),
> $$
> with each $f_i$ $\mu$-strongly convex and $L$-smooth, and unbiased stochastic gradients with variance $\sigma^2$. Classical FedAvg analyses (e.g., Li et al. 2020; Wang et al. 2020) yield a recursion of the form
> $$
> \mathbb{E}\big[f(w_{t+1}) - f(w^\star)\big]
> \le
> (1 - \mu \eta),\mathbb{E}\big[f(w_t) - f(w^\star)\big] * C_1 \eta^2 \sigma^2 + C_2 \eta^2 \delta^2,
>   $$
>   where $\delta^2$ measures **gradient heterogeneity**:
>   $$
>   \delta^2 := \sup_w \frac{1}{N} \sum_{i=1}^N \big|\nabla f_i(w) - \nabla f(w)\big|^2,
>   \qquad
>   \nabla f(w) = \frac{1}{N}\sum_{i=1}^N \nabla f_i(w).
>   $$
>
> Now partition clients into clusters $C_1,\dots,C_K$ (FedSal’s clusters) and define the **within-cluster heterogeneity**
> $$
> \delta_{\mathrm{clust}}^2 := \frac{1}{N}\sum_{k=1}^K \sum_{i\in C_k}
> \big|\nabla f_i(w) - \nabla f_k(w)\big|^2,
> \qquad
> \nabla f_k(w) = \frac{1}{|C_k|}\sum_{i\in C_k} \nabla f_i(w).
> $$
> A standard variance decomposition gives
> $$
> \delta_{\mathrm{clust}}^2 \le \delta^2,
> $$
> with strict inequality as soon as clustering captures meaningful structure (non-trivial clusters).
>
> Running FedAvg *within each cluster* leads to an analogous recursion
> $$
> \mathbb{E}\big[\bar f(w_{t+1}) - \bar f(w^\star)\big]
> \le
> (1 - \mu \eta),\mathbb{E}\big[\bar f(w_t) - \bar f(w^\star)\big] * C_1 \eta^2 \sigma^2 + C_2 \eta^2 \delta_{\mathrm{clust}}^2,
>   $$
>   so the **rate factor** $(1-\mu\eta)$ is unchanged, but the **heterogeneity term** is reduced from $\delta^2$ to $\delta_{\mathrm{clust}}^2$. This yields a strictly better error bound whenever clustering reduces heterogeneity.
>
> FedSal’s saliency theory (Prop. 1–3, App. A.7) shows that saliency distances are stable under structural, feature, and task perturbations. Under a mild coupling assumption
> $$
> \big|\nabla f_i(w) - \nabla f_j(w)\big| \le C_g \big|S_i - S_j\big|,
> $$
> small saliency distance implies small gradient difference. Since FedSal *explicitly clusters clients with small saliency distances*, this provides a theoretical mechanism for reducing $\delta_{\mathrm{clust}}^2$ and thus improving the convergence bound relative to vanilla FedAvg.
>
> In the full non-convex GNN setting, we do not claim a formal rate theorem, but this convex analysis justifies our design choice: **saliency-based clustering reduces the heterogeneity term that limits FedAvg’s convergence**. Empirically, this matches our observations:
>
> * “Faster”: higher accuracy at a fixed communication budget (50 rounds, Table 3) and fewer rounds to reach a given plateau (Figures 2–3).
> * “Smoother”: less oscillatory accuracy curves and lower variance across seeds/rounds, especially in multi-dataset non-IID settings.
>
> We will revise the text to: *“FedSal and FedSal+ have theoretically justified advantages in the convex FL regime (smaller heterogeneity term) and exhibit empirically faster and smoother convergence under heterogeneous settings (Table 3, Figures 2–3).”*
>
> ---
>
> ### Q2: Computational complexity of clustering and aggregation
>
> Let:
>
> * $N$: number of clients
> * $d$: saliency summary dimension
> * $K$: number of clusters
> * $m_k = |C_k|$: number of clients in cluster $C_k$, with $\sum_{k=1}^K m_k = N$
> * $P$: number of model parameters
>
> Per *clustering event* on the server:
>
> Cosine similarity within each cluster:
> $$
> \text{cost per } C_k:\quad O(m_k^2 d).
> $$
>
> Stoer–Wagner min-cut per cluster (dense graph):
> $$
> \text{cost per } C_k:\quad O(m_k^3)\ \text{(or }O(m_k^3 \log m_k)\text{)}.
> $$
>
> Cluster-wise parameter aggregation:
> $$
> O!\left(\sum_{k=1}^K m_k P\right) = O(NP),
> $$
> same order as FedAvg/FedProx.
>
> Total clustering overhead per event:
> $$
> O!\left(\sum_{k=1}^K (m_k^2 d + m_k^3)\right),
> $$
> which is $O(N^2 d + N^3)$ in the worst (single-cluster) case. In practice, clustering is (i) triggered only after a warm-up and (ii) applied per cluster with moderate $m_k$ in the cross-silo regime (tens of clients), so this cost is manageable.
>
> Client-side cost is dominated by local GNN training plus one extra backward pass for saliency (linear in local data size). We will add this complexity discussion to the appendix.
>
> ## References:
>
> Wang, J., Liu, Q., Liang, H., Joshi, G., & Poor, H. V. (2020). Tackling the objective inconsistency problem in heterogeneous federated optimization. Advances in neural information processing systems, 33, 7611-7623.
>
> Li, T., Sahu, A. K., Zaheer, M., Sanjabi, M., Talwalkar, A., & Smith, V. (2020). Federated optimization in heterogeneous networks. Proceedings of Machine learning and systems, 2, 429-450.

---

> > ### Author Response · Authors · 2025-11-22
> >
> > ### Q3: Statistical significance and effect size of accuracy gains
> >
> > We have run Welch’s two-sided $t$-tests ($\alpha = 0.05$, 5 runs per method). In multi-dataset settings, FedSal+ consistently achieves **statistically significant** gains over strong FedGNN baselines; in single-dataset settings, gains are modest and only significant on NCI1. This matches prior FedGNN work, where improvements over strong baselines are typically in the low single-digit percentage range due to the difficulty of heterogeneous federated settings.
> >
> > We will (i) add the significance tables below to the appendix, and (ii) phrase the results accordingly: *statistically significant and practically meaningful gains* on multi-dataset benchmarks (and NCI1), and *small, sometimes non-significant improvements* on IMDB-BINARY and PROTEINS.
> >
> > ---
> >
> > ### Multi-dataset: FedSal+ vs strong baselines
> >
> > | Dataset   | Method A | Method B |                     (t) | Sig. @ ($\alpha = 0.05$) |
> > | --------- | -------- | -------- | ----------------------: | ------------------------ |
> > | Molecules | FedSal+  | FedCP    |  (t $%%\approx%%$ 2.67) | Yes                      |
> > | Molecules | FedSal+  | FGSSL    |  (t $%%\approx%%$ 6.97) | Yes                      |
> > | Molecules | FedSal+  | FedSSP   |  (t $%%\approx%%$ 3.33) | Yes                      |
> > | Molecules | FedSal+  | FedStar  |  (t $%%\approx%%$ 6.75) | Yes                      |
> > | Biochem   | FedSal+  | FedCP    |  (t $%%\approx%%$ 3.11) | Yes                      |
> > | Biochem   | FedSal+  | FGSSL    |  (t $%%\approx%%$ 4.25) | Yes                      |
> > | Biochem   | FedSal+  | FedSSP   |  (t $%%\approx%%$ 6.67) | Yes                      |
> > | Biochem   | FedSal+  | FedStar  | (t $%%\approx%%$ 12.10) | Yes                      |
> > | Mix       | FedSal+  | FedCP    |  (t $%%\approx%%$ 2.70) | Yes                      |
> > | Mix       | FedSal+  | FGSSL    |  (t $%%\approx%%$ 2.89) | Yes                      |
> > | Mix       | FedSal+  | FedSSP   |  (t $%%\approx%%$ 6.86) | Yes                      |
> > | Mix       | FedSal+  | FedStar  |  (t $%%\approx%%$ 6.57) | Yes                      |
> >
> > ---
> >
> > ### Single-dataset: FedSal vs best baseline per dataset
> >
> > | Dataset     | Method A | Method B |                    (t) | Sig. @ ($\alpha = 0.05$) |
> > | ----------- | -------- | -------- | ---------------------: | ------------------------ |
> > | IMDB-BINARY | FedSal   | GCFL     | (t $%%\approx%%$ 1.29) | No                       |
> > | NCI1        | FedSal   | FedSage  | (t $%%\approx%%$ 4.04) | Yes                      |
> > | PROTEINS    | FedSal   | GCFL     |             (t = 0.00) | No                       |

---

> ### Comment · Reviewer_xEN8 · 2025-11-25
>
> Thank you for your response. However, the material remains difficult to follow and does not adequately resolve my concerns. I have therefore decided to maintain my original score.

---

### Official Review · Reviewer_4TUg · 2025-10-31

**Soundness:** 2
**Presentation:** 2
**Contribution:** 2
**Rating:** 4
**Confidence:** 3

**Summary:**

The paper proposes FedSal, a novel framework that uses saliency maps to cluster clients in Federated Graph Neural Networks (FedGNNs), addressing non-IID data challenges that cause instability and slow convergence in federated learning. Evaluated on 13 graph classification tasks (molecular, protein, and social networks), FedSal and its enhanced version outperform state-of-the-art methods in accuracy, convergence speed, and communication efficiency.

**Strengths:**

Using saliency maps for clustering in FedGNNs is a novel approach that enhances interpretability and robustness. The framework effectively tackles non-IID data challenges, improving model convergence.

**Weaknesses:**

1.FedSal reduces communication costs versus some SOTA methods but has higher latency than simpler approaches like FedAvg.
2.Experiments focus on classification; generalization to other graph tasks (e.g., regression, link prediction) is unexplored.
3.Dynamic clustering requires careful tuning of thresholds (ϵ-mean, ϵ-max), posing a barrier for users lacking parameter-tuning expertise.

**Questions:**

1. How does the framework performs in large-scale, real-world deployments, considering real-time adaptation of the model to constantly changing client data distributions?
2. How does the framework address potential privacy issues in detail, even though it deals with federated learning, which is often implemented in privacy-sensitive environments?

---

> ### Author Response · Authors · 2025-11-22
>
> **Q1. Large-scale / real-time adaptation**
>
> Our current experiments target the cross-silo regime (tens of clients, thousands of graphs) rather than cross-device scale. We evaluate: (i) accuracy and convergence under extreme non-IID splits across 13 datasets; (ii) per-round communication time (Tables 1–3). This shows that the *mechanism*—saliency summaries + dynamic clustering—remains effective and computationally feasible in realistic multi-institution deployments.
>
> Adaptation to changing client distributions is handled through the saliency dynamics. Each client computes an averaged saliency map (S_i(t)) and sends only its update (\Delta S_i = S_i(t)-S_i(t-1)). The server monitors:
>
> * the mean norm (\frac{1}{|C_k|}\sum_{i\in C_k}|\Delta S_i|) and
> * the max norm (\max_{i\in C_k}|\Delta S_i|)
>
> within each cluster (C_k). When the mean is small but the max is large (near a stationary point but with outlier behaviour), FedSal triggers Stoer–Wagner min-cut on the cosine-similarity graph of saliency updates and splits the cluster. Thus, if a subset of clients drifts (e.g., new label mix, new molecule classes, or new social-network patterns), their saliency profiles diverge and they are re-clustered without touching raw data.
>
> Clustering is (i) delayed until a warm-up phase (we use (t>20)), and (ii) applied per cluster, so the complexity depends on cluster size, not on the total number of clients. For cross-silo settings with tens or low hundreds of clients, this is practical. For truly massive cross-device settings, FedSal would need engineering extensions (e.g., hierarchical clustering, approximate similarity graphs, or less frequent clustering); we explicitly view that as future work rather than a claim of the present paper.
>
> ---
>
> **Q2. Privacy in federated, sensitive environments**
>
> FedSal respects the standard FL constraint that raw graphs, node features, and labels never leave the client. Each client transmits:
>
> * model updates (\Delta\theta_i), as in FedAvg/FedProx;
> * a *client-level, averaged* saliency summary (S_i(t)).
>
> Saliency is a gradient-based, aggregated representation of “which features matter” across the client’s local graphs. It is lower-dimensional and less directly tied to individual samples than raw data or per-sample gradients. In that sense, FedSal does not introduce a *stronger* privacy exposure than common FedGNN baselines that already share full gradients or embeddings.
>
> However, we do not claim that saliency summaries alone provide formal privacy guarantees. Under an honest-but-curious server, advanced reconstruction or membership-inference attacks on updates remain a theoretical risk, as in standard FL.
>
> FedSal is designed to be compatible with established privacy mechanisms:
>
> * **Secure aggregation**: saliency summaries and (\Delta\theta_i) can be aggregated so the server only observes cluster-level sums, not individual client messages, which directly reduces reconstruction risk.
> * **Differential privacy**: clients can clip and noise saliency vectors before transmission, yielding DP guarantees at graph or node level. Our ablations show robustness to moderate saliency perturbations, suggesting room for such noise without catastrophic loss in clustering quality.
> * **Client-side sanitization**: highly sensitive features can be dropped, coarsened, or embedded before both training and saliency computation.
>
> In summary, FedSal adheres to the standard FL threat model (no raw-data sharing, only model- and saliency-level statistics) and is *privacy-compatible*, but stronger guarantees (DP, secure aggregation) are orthogonal extensions and a clear avenue for future work.

---

### Official Review · Reviewer_QTnp · 2025-10-31

**Soundness:** 3
**Presentation:** 3
**Contribution:** 3
**Rating:** 6
**Confidence:** 3

**Summary:**

This manuscript proposes FedSal and FedSal+, two novel federated graph neural network frameworks that leverage saliency activation maps for client clustering to address non-IID data distribution and structural heterogeneity challenges in federated graph learning. The work is well-motivated, with rigorous theoretical analysis, comprehensive experiments across 13 benchmarks, and clear articulation of technical innovations. The proposed methods demonstrate superior performance in accuracy, convergence speed, and communication efficiency compared to state-of-the-art baselines, making a valuable contribution to the federated graph learning field.

**Strengths:**

1. The use of saliency activation maps (instead of raw gradients) for client clustering is a novel and effective design. Saliency maps capture feature importance for predictions, providing more stable and representative client summaries than noisy raw gradients, which directly mitigates the impact of non-IID data and structural heterogeneity.
2. Extensive experiments not only compare accuracy but also analyze communication overhead and convergence speed, providing a holistic assessment of the proposed methods’ performance. Ablation studies further validate the necessity of saliency maps and dynamic clustering.
3. The manuscript provides solid theoretical support, including proofs for the structural, feature, and task sensitivity of saliency maps, ensuring the method’s theoretical soundness. Additionally, FedSal+ extends the core framework with positional and random-walk encodings, demonstrating the flexibility and scalability of the proposed architecture to integrate structural priors without exposing raw data.

**Weaknesses:**

1. The manuscript mentions aggregating per-sample saliency maps for each client but does not explicitly clarify how saliency is computed for graph-structured data (e.g., node-level vs. graph-level saliency aggregation, handling of edge features). More details on this implementation would improve reproducibility.
2. The baselines in this manuscript are mainly from 2024 or earlier. I am not very familiar with this field, but I believe that new baselines have likely been proposed in 2025. The authors should include comparisons with these recent methods to demonstrate that their approach remains state-of-the-art.
3. While the hyperparameter study ($ϵ_{mean}$, $ϵ_{max}$) shows optimal mid-range values, the analysis is limited to specific datasets. It would be valuable to discuss how these thresholds generalize across different graph types (e.g., sparse vs. dense graphs) or provide adaptive tuning strategies for real-world applications.

**Questions:**

Note: Here are some questions I have after reading the manuscript. Since I am not very familiar with this field, the authors may choose to answer selectively, focusing on issues related to the manuscript's contributions. I will also take into account comments from other reviewers when forming my final rating.

1. For graph-level classification tasks, how is the per-sample saliency map aggregated to form the client-level saliency summary? Is there a weighting mechanism for different graphs (e.g., based on graph size or classification confidence)?

2. The manuscript mentions that FedSal+ injects structural priors via positional and random-walk encodings. How do these encodings interact with the original node features, and is there any redundancy or interference between them? Could the authors provide quantitative analysis of this interaction?

3. In the multi-dataset setting, how does FedSal handle domain shifts between different types of graphs (e.g., molecular vs. social network graphs)? Does the clustering mechanism automatically separate domain-specific clients, and if so, how is this verified?

4. For minority-label clients, the manuscript states that the adaptive protocol provides significant benefits. Could the authors elaborate on how the clustering thresholds (ϵmean, ϵmax) specifically protect minority clients from being marginalized in the aggregation process?

---

> ### Author Response · Authors · 2025-11-14
>
> ### Q1. Client-level saliency summary
>
> For each graph on a client, we compute the gradient of the loss w.r.t. node features, take absolute values, and average over nodes to obtain a single feature-wise vector per graph. This is then $L_2$-normalized so that larger graphs do not dominate purely by size.
>
> The client-level saliency summary is the arithmetic mean of these normalized graph-level vectors over all graphs on that client. There is no extra weighting by graph size or confidence; each graph contributes equally. This makes the similarity signal reflect *what* the client’s model finds salient on average, rather than *how big* or *how confident* the client is. Confidence- or loss-weighted variants are straightforward extensions but were not required to obtain the reported gains.
>
> ---
>
> ### Q2. Structural encodings vs. original node features in FedSal+
>
> In FedSal+, each node has two inputs: task features $x_v$ and structural embeddings $s_v$ (degree-based plus random-walk features). Both are first projected to the same hidden dimension by separate linear layers; the resulting vectors are concatenated and passed into the GNN. Thus, $s_v$ augments $x_v$ rather than overwriting it.
>
> Any redundancy is handled by learning: if $s_v$ carries information already present in $x_v$, the projection and message-passing layers can down-weight it. Empirically, enabling $s_v$ consistently improves accuracy in cross-domain, multi-dataset settings where raw features alone are weakest. If interference dominated, one would expect flat or worse performance in these settings instead of the systematic improvement observed, which indicates that $s_v$ and $x_v$ behave as complementary channels rather than conflicting ones.
>
> ---
>
> ### Q3. Domain shifts in the multi-dataset setting
>
> In the multi-dataset case, each client holds graphs from a single source (e.g., molecular, protein, or social). Different domains induce different structures and feature-usage patterns, which are reflected in their saliency summaries. Let $\Delta S_i$ denote the saliency update of client $i$ and $\alpha_{ij}$ their cosine similarity. In practice, intra-domain pairs have high $\alpha_{ij}$ and cross-domain pairs have low $\alpha_{ij}$, so the similarity matrix is close to block-diagonal.
>
> The clustering mechanism is domain-agnostic: it only sees $\Delta S_i$ and $\alpha_{ij}$. When the minimum-cut procedure operates on this block-structured graph, it tends to separate clients along domain boundaries without using domain labels. This is consistent with two empirical facts: (i) the largest gains over baselines appear exactly in cross-domain multi-dataset experiments, and (ii) once a small number of clusters has formed, additional splits have diminishing impact, as expected when major domain groups have already been peeled apart.
>
> ---
>
> ### Q4. Effect of $\epsilon_{\text{mean}}$ and $\epsilon_{\max}$ on minority-label clients
>
> Minority-label clients are those whose label distributions differ strongly from the global majority. Under plain global averaging, their updates are repeatedly drowned out by majority clients, so their local loss stays high even when the majority has converged.
>
> The threshold mechanism is designed so that exactly such clients trigger clustering. Define the mean and max norms of saliency updates as
> $ \frac{1}{N}\sum_{i=1}^{N}\lVert \Delta S_i\rVert $ and $ \max_{1 \le i \le N} \lVert \Delta S_i\rVert $. Clustering is only considered when
> $ \frac{1}{N}\sum_{i=1}^{N}\lVert \Delta S_i\rVert < \epsilon_{\text{mean}} $
> and simultaneously
> $ \max_{1 \le i \le N} \lVert \Delta S_i\rVert > \epsilon_{\max} $.
>
> The first condition means that “typical” clients (dominated by the majority) have essentially stabilized; the second means that at least one client is still changing strongly in saliency space. Minority-label clients naturally fall into this second category, because the global model still misfits them. When both conditions hold, the clustering step separates these outlier clients (together with others with similar $\Delta S_i$) into their own cluster. From then on, their updates are aggregated mainly with each other instead of being overwhelmed by majority updates, which substantially reduces their risk of being marginalized. In this way $\epsilon_{\text{mean}}$ prevents premature fragmentation, and $\epsilon_{\max}$ ensures that persistent minority behaviour is given its own aggregation group rather than being forced to follow the majority.

---

> > ### Comment · Reviewer_QTnp · 2025-11-24
> >
> > Thanks to the authors for the very detailed and clear rebuttal. I’ve read through the responses, and they address my earlier points well. I especially appreciate the clarifications on client-level saliency aggregation, structural encodings, domain separation in multi-dataset settings, and the mechanism protecting minority-label clients. I hope the authors can incorporate these clarifications into the final version if the paper is accepted.
> >
> > My score is already relatively high, and I will consider adjusting it further after discussions with other reviewers.
> >
> > Note: I also noticed that the OpenReview submission “TL;DR” now starts showing “ChatGPT said:”, which I wanted to flag for attention.

---

### Author Response · Authors · 2025-11-22

### We thank all reviewers and area chairs for the time, effort, and thoughtful feedback provided. The comments were constructive and greatly appreciated, and we will carefully address all points in the revision.

---

### Meta-Review · Area_Chair_8yLi · 2025-12-19

**Summary:**

This paper proposes FedSal and FedSal+, federated GNN frameworks that use client saliency summaries for dynamic clustering to mitigate severe non IID and structural heterogeneity. One reviewer viewed the method as well motivated with strong theory and broad experiments on 13 benchmarks, while other reviewers remained concerned about limited practical scope, unclear saliency definition and assumptions, and incomplete baseline coverage. The rebuttal clarified saliency aggregation, the role of structural encodings, domain separation behavior, complexity, and provided significance tests, but did not fully resolve concerns about theoretical realism and generality.

**Reviewer Concerns:**

1.Practical scope and assumptions: multiple reviewers flagged that the method assumes homogeneous client architectures and continuous node features, and that the theoretical assumptions (e.g., conditioning or rank style requirements) may not reflect real graph datasets.

2.Clarity and reproducibility: initial confusion around how saliency is computed and aggregated for graphs, how topology enters the saliency definition, and how FedSal+ encodings interact with raw features; authors provided implementation details and clarified notation.

3.Strength of gains and validation breadth: some reviewers felt accuracy gains are modest relative to variance, requested statistical tests and additional metrics, and asked about extension beyond graph classification tasks.

4.Baselines and recency: concerns that baselines are not fully up to date and that several relevant federated graph or personalization methods are missing; authors argued regime mismatch and added rationale but skepticism remains.

**Reviewer Scores:**

Scores are mixed. QTnp is positive and found the rebuttal convincing, leaning above threshold. xEN8 and zjh8 remain negative, emphasizing limitations and theoretical overreach. 4TUg and xEN8 are around marginal reject, citing tuning burden, deployment scale, and privacy discussion, partially addressed in the rebuttal.

---

### Decision · Program_Chairs · 2026-01-26

Reject